# Hedgehog signaling is a potent regulator of liver lipid metabolism and reveals a GLI-code associated with steatosis

Madlen Matz-Soja[1]*, Christiane Rennert[1], Kristin Schönefeld[1], Susanne Aleithe[1], Jan Boettger[1], Wolfgang Schmidt-Heck[2], Thomas S Weiss[3], Amalya Hovhannisyan[1†], Sebastian Zellmer[1], Nora Klöting[4], Angela Schulz[1], Jürgen Kratzsch[5], Reinhardt Guthke[2], Rolf Gebhardt[1]

[1]Institute of Biochemistry, Faculty of Medicine, Leipzig University, Leipzig, Germany; [2]Leibniz Institute for Natural Product Research and Infection Biology – Hans Knöll Institute, Jena, Germany; [3]University Children Hospital, Regensburg University Hospital, Regensburg, Germany; [4]Integrated Research and Treatment Centre Adiposity Diseases, Faculty of Medicine, Leipzig University, Leipzig, Germany; [5]Institute of Laboratory Medicine, Clinical Chemistry and Molecular Diagnostics, Leipzig University, Leipzig, Germany

*For correspondence: madlen.matz@medizin.uni-leipzig.de

Present address: [†]Westpfalz-Clinical Center GmbH, Kaiserslautern, Germany

**Abstract** Non-alcoholic fatty liver disease (NAFLD) is the most common liver disease in industrialized countries and is increasing in prevalence. The pathomechanisms, however, are poorly understood. This study assessed the unexpected role of the Hedgehog pathway in adult liver lipid metabolism. Using transgenic mice with conditional hepatocyte-specific deletion of Smoothened in adult mice, we showed that hepatocellular inhibition of Hedgehog signaling leads to steatosis by altering the abundance of the transcription factors GLI1 and GLI3. This steatotic 'Gli-code' caused the modulation of a complex network of lipogenic transcription factors and enzymes, including SREBP1 and PNPLA3, as demonstrated by microarray analysis and siRNA experiments and could be confirmed in other steatotic mouse models as well as in steatotic human livers. Conversely, activation of the Hedgehog pathway reversed the "Gli-code" and mitigated hepatic steatosis. Collectively, our results reveal that dysfunctions in the Hedgehog pathway play an important role in hepatic steatosis and beyond.

## Introduction

As the most common liver disease in the western countries and a disease with an increasing prevalence, non-alcoholic fatty liver disease (NAFLD) is the subject of many investigations and studies (*Ahmed et al., 2015*; *Dongiovanni et al., 2015*; *Zhang and Lu, 2015*) to identify unknown risk factors and new treatment strategies. Hepatic steatosis is the hallmark feature of NAFLD (*Enomoto et al., 2015*) and has the potential to develop into more severe steatohepatitis (NASH), which can progress to fibrosis, cirrhosis and cancer. Steatosis occurs when the rate of fatty acid delivery exceeds the rate of fatty acid removal (oxidation and export). A comprehensive knowledge of the hepatic lipid metabolism and its control mechanisms is crucial for preventing and treating liver steatosis. To date, these control mechanisms, e.g. the factors governing hepatocyte heterogeneity in lipid metabolism, remain largely unknown and have not yet received adequate attention in the discussion of NAFLD (*Postic and Girard, 2008*). Metabolic zonation of the liver is of considerable importance for the optimal integration and regulation of the plethora of different hepatic functions and metabolic homeostasis (*Gebhardt, 1992*; *Gebhardt and Matz-Soja, 2014*). Recently, Wnt/beta-

**eLife digest** The liver is one of the main organs responsible for processing everything that mammals eat and drink. Nutrients absorbed by the gut like sugars and lipids (fats) are processed by the liver and are stored or distributed to provide energy to other organs. Sometimes these metabolic processes become unbalanced. This can lead to lipids accumulating in the liver – a process known as steatosis, which is a feature of human non-alcoholic fatty liver disease.

In organs like the liver, cells are instructed how to behave via signaling pathways. A protein outside the cell signals to specific proteins inside, which switch on a set of target genes. One such pathway is the Hedgehog pathway, which primarily regulates tissue regeneration and the development of embryos. A component of this pathway is the Smoothened gene, which indirectly switches on proteins called GLI factors that regulate metabolic genes, including those involved in lipid metabolism. The Hedgehog pathway has been found to control the metabolism of lipids in fat tissue but it is not known whether it is important for lipid metabolism in the liver.

Matz-Soja et al. investigated this possible role of the Hedgehog pathway in the liver using mice with a Smoothened gene that could be deleted specifically in that organ. This deletion disrupted Hedgehog signaling and led to lipids accumulating in the liver and eventually to steatosis. These changes were associated with an increase in the amounts and activityof several enzymes (and the proteins that regulate these enzymes) that help to synthesize lipids. Steatosis was also associated with low amounts of two of the three GLI factors; indeed, this seems to be key for triggering problems with lipid metabolism. Human livers with steatosis showed the same changes in levels of the GLI factors.

Increasing the amount of GLI factors in liver cells taken from mice with steatosis reduced the accumulation of lipids and brought lipid metabolism back to its normal balance. A focus of future studies will be to understand how the Hedgehog signaling pathway interacts with other signaling pathways known to regulate liver lipid metabolism, such as insulin signaling. This knowledge will help clinicians to design new treatments for lipid-associated diseases like non-alcoholic fatty liver disease.

catenin signaling was recognized as a master regulator of the zonal distribution of nitrogen metabolism in the adult liver (*Benhamouche et al., 2006*; *Gebhardt and Hovhannisyan, 2010*; *Monga, 2015*) and it influences the balance between the anabolic and catabolic functions of glucose metabolism (*Chafey et al., 2009*). The Hedgehog (Hh) signaling cascade is another important pathway that determines embryonic patterning, cell growth and tissue repair (*Omenetti et al., 2011*; *Gu and Xie, 2015*), and often acts in close crosstalk with Wnt/beta-catenin signaling (*Toku et al., 2011*). Similar to Wnt/beta-catenin signaling, the Hh pathway can act in canonical and non-canonical manners, as explicitly described by Teperino et al. (*Teperino et al., 2014*). To activate canonical Hh signaling in mammals, the ligands Sonic, Indian and Desert Hedgehog (SHH, IHH and DHH) interact with the surface receptors Patched 1 and 2 (PTCH1 and PTCH2) which remove their inhibition of the co-receptor Smoothened (SMO). Active SMO triggers the activation of the GLI (Glioma-associated oncogene) transcription factors (TFs) (GLI1, 2 and 3) by preventing the conversion of GLI2 and GLI3 into transcriptional repressors (*Ruiz i Altaba, 1999*; *Infante et al., 2015*). According to Ruiz I Altaba et al., the 'Gli-code', i.e. the combinatorial and cooperative function of the repressing and activating forms of all GLI factors, forms the basis of the integration of Hh signals (as well as of multiple other morphogens and cytokines) in embryogenesis and carcinogenesis (*Ruiz i Altaba, 1999*; *Ruiz i Altaba et al., 2003*; *2007*). To date, Hh signaling has been found to play a considerable role in various scenarios of adult liver regeneration from progenitor cells (*Sicklick et al., 2006*) and in the regulation of the compensatory outgrowth of progenitors and myofibroblasts (*Jung et al., 2010*). Regarding liver metabolism, we recently showed that the regulation of the IGF-axis (insulin-like growth factor) in mature hepatocytes is controlled by Hh signaling (*Matz-Soja et al., 2014*). Moreover, we were also able to show that the TFs GLI1, GLI2 and GLI3 form a unique self-stabilizing network in hepatocytes and regulate several metabolic genes, including lipid associated factors (*Schmidt-Heck et al., 2015*). Therefore, in this study, we have focused on the alterations in lipid metabolism as a consequence of

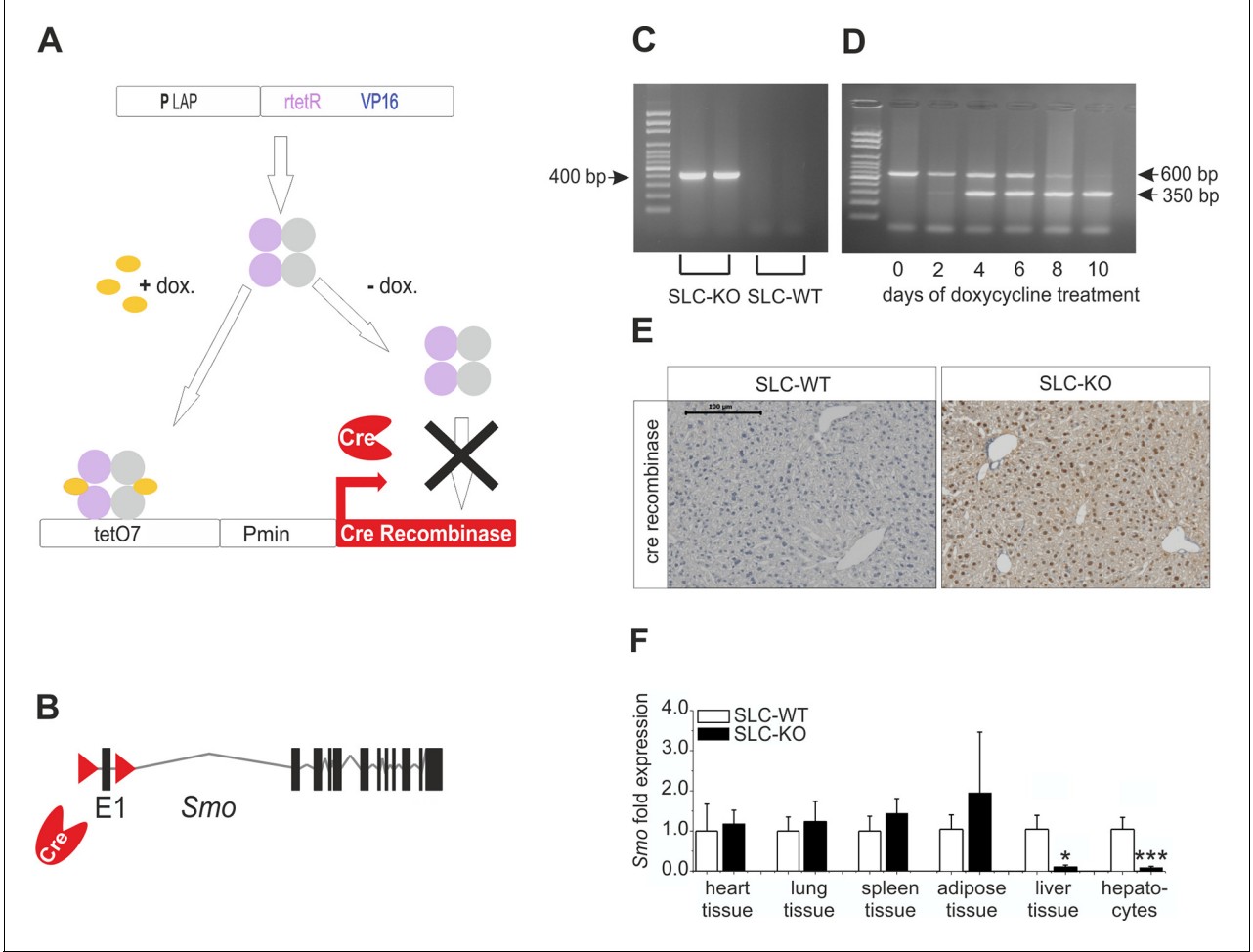

**Figure 1.** Strategy for conditional and hepatocyte-specific deletion of *Smo*. (**A**) Scheme of the tet-on system in the LC-1/rTALAP-1 mice. (**B**) Structure of *Smo* with loxP sites. (**C**) PCR, for Cre recombinase, yielded a 400 bp fragment in SLC-KO mice only. (**D**) PCR from liver tissue during treatment with doxycycline, yielding a 600 bp amplicon for SLC-WT alleles and a 350 bp amplicon for the recombinant *Smo* alleles in the SLC-KO mice. (**E**) Immunohistochemical staining of Cre recombinase in liver sections of the SLC-WT and the SLC-KO mice. Bar: 100 µm. (**F**) qRT-PCR of *Smo* in different tissues and isolated hepatocytes of the SLC-WT (n=6–10) and the SLC-KO (n=6–10) mice. Source files of all data used for the quantitative analysis are available in the *Figure 1—source data 1*.

The following source data and figure supplement are available for figure 1:

**Source data 1.** Source data of qRT-PCR of *Smo* in different tissues and isolated hepatocytes of the SLC-WT and the SLC-KO mice (*Figure 1F*).

**Figure supplement 1.** Influence of doxycycline on lipogenic gene expression qRT-PCR of *Ppara*, *Pparb/d*, *Pparg*, *Elovl3*, *Elovl6* and *Fasn* in isolated hepatocytes of the male SLC-WT mice treated with doxycycline (n=3–4) for 10 days compared to male SLC-WT mice without doxycycline (n=3–4) at the same age.

hepatic Hh signaling modulations in vivo and in vitro. To address this aim, a mouse model with a conditional hepatocyte-specific deletion of *Smo* in adult mice was established, in order to avoid interference with developmental effects of Hh signaling. For the in vitro investigations, we used RNA interference (RNAi) technology to knock down several genes of the Hh pathway in cultured hepatocytes (*Böttger et al., 2015*; *Schmidt-Heck et al., 2015*). The results of our investigations clearly show that the Hh pathway is a strong regulator of lipid metabolism in the adult liver. Furthermore, we show that impaired Hh signaling leads to increased expression of lipogenic TFs and enzymes with different zonal preference which finally results in steatosis. Conversely, we demonstrate that

slight activation of the pathway by Sufu knockdown, small molecule agonists or GLI overexpression can mitigate lipid accumulation in steatotic livers.

## Results

### Breeding of transgenic mice

As depicted in *Figure 1A–B*, triple transgenic mice which have a conditional hepatocyte-specific ablation of *Smo* in response to doxycycline (abbreviated SLC mice) were generated. Thus, the Smo<sup>t-m2Amc</sup>/J mice (Jackson Laboratories), which possess *loxP* sites flanking exon 1 of the *Smo* gene (*Long et al., 2001*), were crossed with the LC-1/rTA<sup>LAP</sup>-1 mice which are working with a tetracycline-controlled transcriptional activation of the Cre recombinase protein (provided by Hermann Bujard) (*Schonig et al., 2002*). In the LC-1/rTA<sup>LAP</sup>-1 mice the synthetic transactivator variant (rtetR) of the tet-repressor present in rTALAP-1 mice is driven by the LAP-promoter (PLAP). In the presence of doxyxycline, rtetR binds to an array of seven tet operator sequences (tetO7) activating transcription of the Cre recombinase gene (tet-on-system) (*Figure 1A*). The offspring were genotyped by PCR for the *Smo* wildtype (*Smo* WT) and *Smo* floxed (*Smo* flx) alleles, the doxycycline responsible element (*rtetR*) and the Cre recombinase (Primer: *Supplementary file 1A*).

At the age of 8 weeks (to avoid the hormonal complications of adolescence), hepatocyte-specific ablation of *Smo* was induced by adding 2 mg/ml doxycycline hydrochloride (Sigma, Germany) to the drinking water for 10 days to promote the expression of the Cre recombinase (*Figure 1C*). During this period, the *Smo* rec. (recombinant) primer yielded a 350 bp fragment, indicating the deletion of the floxed domain (*Figure 1D*) (Primer: *Supplementary file 1A*). After 10 days, nearly all hepatocytes were positive for Cre recombinase protein (*Figure 1E*). After this treatment, the mice were maintained under normal conditions until 12 weeks of age. After sacrifice, the specificity of the *Smo* deletion was measured via qRT-PCR, indicating that there was a significant decrease in *Smo* expression in the liver material and isolated hepatocytes (*Figure 1F*). As already shown in our previous article, no adverse side effect of the doxycycline treatment could be observed on body weight (*Matz-Soja et al., 2014*). Likewise, the expression of important genes of lipid metabolism in hepatocytes was not affected by doxycycline (*Figure 1—figure supplement 1*).

### Phenotypic features of *Smo* knockout mice

When maintained without doxycycline, the transgenic SLC mice developed without a phenotype. The deletion of *Smo* at 8 weeks of age resulted in pronounced liver steatosis within 5 weeks (*Figure 2A*). The H&E and fat red staining clearly showed lipid droplet accumulations, which are most prominent in the midzone to periportal zone, but occasionally encompassed the entire parenchyma (*Figure 2A*). Quantification of the fat red staining revealed a 7-fold increase in the SLC-KO mice compared with the WT controls (*Figure 2B*). Because the insulin and glucose levels changed only marginally (*Table 1*), we conclude that insulin resistance does not contribute to the steatosis in SLC-KO mice at least during the first five weeks after doxycycline administration. This assumption is confirmed by the fact that we could not detect any significant changes in hepatic expression for insulin receptor (*Insr*), insulin receptor substrate 1 and 2 (*Irs1/2*) (*Figure 2—figure supplement 1*). With the exception of the steatosis and lower liver/body weight ratio compared with the SLC-WT mice (*Figure 2C*), there were no other indications of overt liver damage in the SLC-KO mice. Accordingly, the serum activities of ASAT (aspartate aminotransferase), ALAT (alanine aminotransferase) and GLDH (glutamate dehydrogenase) were not different from those of the SLC-WT mice (*Table 1*). The plasma triglyceride levels were significantly increased in the VLDL (very low density lipoprotein) fraction, while no changes were found in LDL (low density lipoprotein) and HDL (high density lipoprotein) fractions (*Figure 2—figure supplement 2*). These findings indicate that steatosis is not caused by an impairment of triglyceride secretion from the hepatocytes.

In addition to these results *in vivo*, the inhibition of Hh signaling in cultured mouse and human hepatocytes using the SMO inhibitor cyclopamine (*Hovhannisyan et al., 2009*) also resulted in marked steatosis within 48 to 72 hr (*Figure 2—figure supplement 3*).

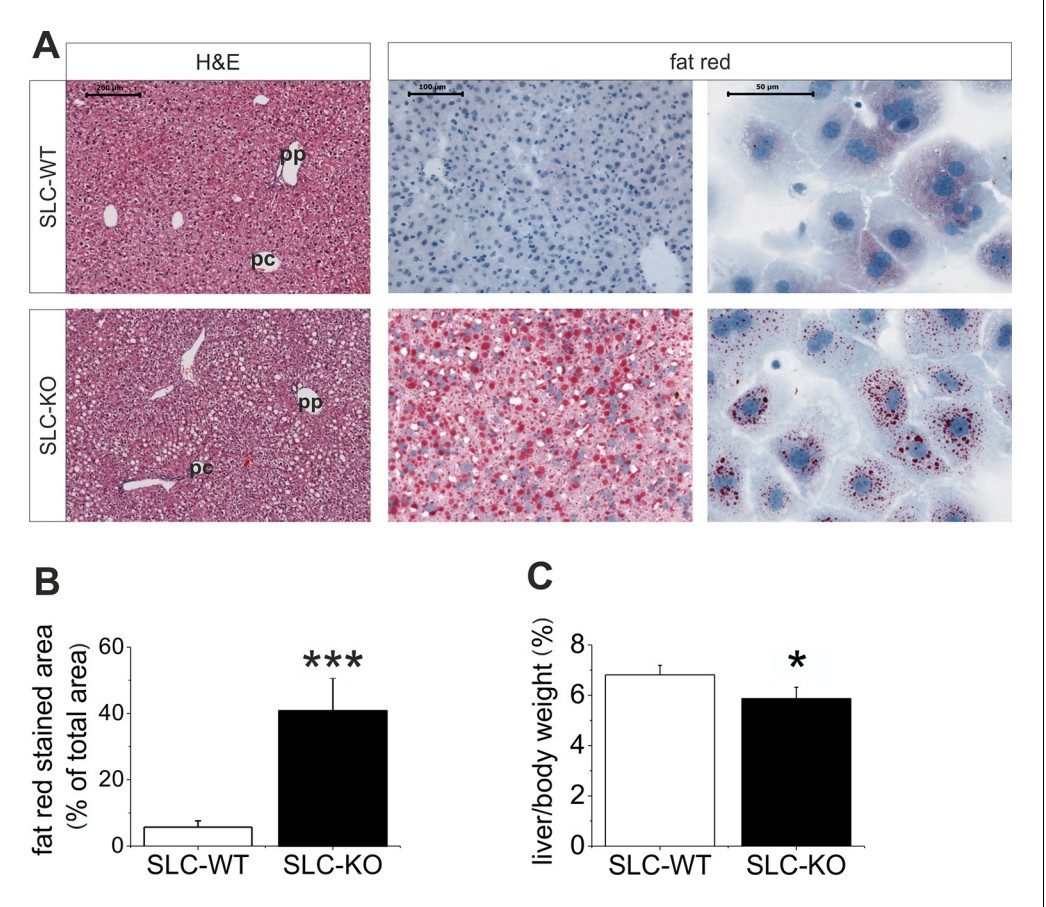

**Figure 2.** Liver phenotype of the SLC mice. (**A**) The H&E and fat red staining of liver sections and hepatocytes showed strong steatosis in the male SLC-KO mice compared to the SLC-WT mice (bars: 200 µm, 100 µm and 50 µm) (pc: pericentral, pc: periportal). (**B**) Quantification of the fat red-stained liver sections from the male SLC-WT (n=10) and SLC-KO (n=7) mice. (**C**) Comparison of the liver/body ratio. Source files of all data used for the quantitative analysis are available in the *Figure 2—source data 1*.

The following source data and figure supplements are available for figure 2:

**Source data 1.** Source data of quantification of the fat red-stained liver sections from the male SLC-WT and SLC-KO mice (*Figure 2B*) and comparison of the liver/body ratio (*Figure 2C*).

**Figure supplement 1.** Gene expression of insulin signaling in isolated hepatocytes from SLC mice.

**Figure supplement 2.** Serum lipoprotein levels of the SLC mice.

**Figure supplement 3.** Influence of cyclopamine on accumulation of neutral lipids in cultured mouse and human hepatocytes.

## Alterations in downstream signaling of the Hh Pathway due to the deletion of *Smo*

To study the signaling cascade after the deletion of *Smo*, we analyzed the alterations of Hh-related genes in isolated hepatocytes. Regarding Hh ligands, we could observe a down-regulation of *Ihh* and *Shh*, consistent with a decrease in pathway activity due to the deletion of *Smo* (*Figure 3A*). It is worth noting that *Ihh* is the most abundantly expressed ligand in hepatocytes, whereas *Shh* is hard to detect and *Dhh* was not measureable. Regarding the receptors, we did not observe significant changes in the *Ptch1, Ptch2* and *Hhip* (Hedgehog-interacting protein) transcripts, which was also

**Table 1.** Serum concentrations of glucose, insulin and enzyme activities of ALAT, ASAT and GLDH in SLC mice.

| parameter | SLC-WT | n | SLC-KO | n |
|---|---|---|---|---|
| insulin (pmol/l) | 80.06 ± 9.98 | 13 | 110.88 ± 31.31 | 7 |
| glucose (mmol/dl) | 10.16 ± 1.80 | 5 | 7.68 ± 1.02 | 6 |
| ALAT (µkat/l) | 1.02 ± 0.32 | 5 | 0.81 ± 0.03 | 5 |
| ASAT (µkat/l) | 2.95 ± 0.90 | 5 | 2.50 ± 0.41 | 5 |
| GLDH (µkat/l) | 0.42 ± 0.13 | 5 | 0.28 ± 0.05 | 5 |

true for *Fu* (Fused) and *Sufu* (Suppressor of fused) (*Figure 3B,C*). However, the *Gli1* and *Gli3* mRNAs were significantly decreased in the SLC-KO mice, while the *Gli2* mRNA remained unchanged (*Figure 3D*). To visualize the amount and the distribution of the GLI1 and GLI3 protein in the liver parenchyma of the SLC mice, we performed immunohistological stainings of GLI1 and GLI3 as well. (*Figure 3E,F*). The results clearly demonstrate that GLI1 and GLI3, are well detectable in hepatocyte nuclei in SLC-WT mice, but are absent in nuclei of SLC-KO hepatocytes (white arrows). In SLC-KO livers, these TFs are only present in non-parenchymal cells, e.g. bile duct epithelial cells (*Figure 3E,F*, yellow arrowheads). These results were confirmed by analyses of the GLI3 protein content by western blotting in isolated hepatocytes from SLC-WT and SLC-KO mice. The results clearly show that the amount of GLI3/A (full length activator protein) was significantly reduced in SLC-KO hepatocytes (*Figure 3—figure supplement 1A,B*). In order to find out whether these distinct alterations of the GLI factors are characteristic for steatotic livers, we measured the expression signature of the *Gli* TFs in isolated hepatocytes from melanocortin-4-receptor-deficient mice (MC4R) and Lep[ob/ob] mice, which are characterized by massive steatosis as a result of over-nutrition (*Sandrock et al., 2009*; *Itoh et al., 2011*; *Trak-Smayra et al., 2011*). In both mouse models, the expression of *Gli1* and *Gli3* was obviously reduced, whereas *Gli2* expression either did not change (MC4R-KO mice) or was increased (*ob/ob* mice) (*Figure 3—figure supplement 2A,B*). Furthermore we could detect the same transcriptional changes of the Gli factors in human patients with clinical relevant steatosis compared to non-steatotic patients (*Figure 3—figure supplement 2C*).

## Gene set enrichment analysis of global gene expression

To get an impression of the global changes in gene expression in response to the deletion of *Smo*, microarray studies were performed. In order to account for possible inter-individual variations of gene expression, total RNA was prepared from freshly isolated hepatocytes of four pairs of SLC-KO and SLC-WT mice. At a cut-off level of 1.5-fold, 179 genes were up-regulated in Chip Arrays and 106 genes were down-regulated. Gene set enrichment analysis (GSEA) using ClueGO revealed highly significant changes in a number of metabolic functions, particularly those involved in lipid metabolism (*Figure 4A*) (*Figure 4—source data 1—data*). For instance, the GO term 'lipid metabolic process' showed pronounced up-regulation of 30 genes (p-value of 7.70E-11) many of which (e.g. *Ppara*, *Pparg*, *Srebf1*, *Aacs*, *Elovl6* and *Fas*) were verified by qRT-PCR. In addition, the GSEA revealed that many regulated genes in the SLC-KO mice (e.g. *Cd36*, *Avpr1a*, *Ttc23*, *Lifr* and *Fabp2*) belong to the top 50 ones described to be mostly correlated with elevated hepatic triglyceride level among 100 unique inbred mouse strains (*Hui et al., 2015*). The GO term 'metabolic process' showed pronounced up-regulation of 113 genes reaching a p-value of 1.24E-9 indicating that genes from other metabolic processes also responded to the *Smo* knockout. The GO terms 'organic acid metabolic process' and 'steroid biosynthetic process' showed up-regulation of several genes with p-values of 1.61E-6 and 6.09E-5, respectively. Furthermore, genes belonging to the GO term 'response to hormone stimulus' such as *Lipin1* and *Ramp2* were found to be up-regulated. Among down-regulated genes those involved in 'activation of protein kinase activity' and 'microtubule-based processes' prevailed with p-values of 1.59E-3 and 1.62E-3, respectively.

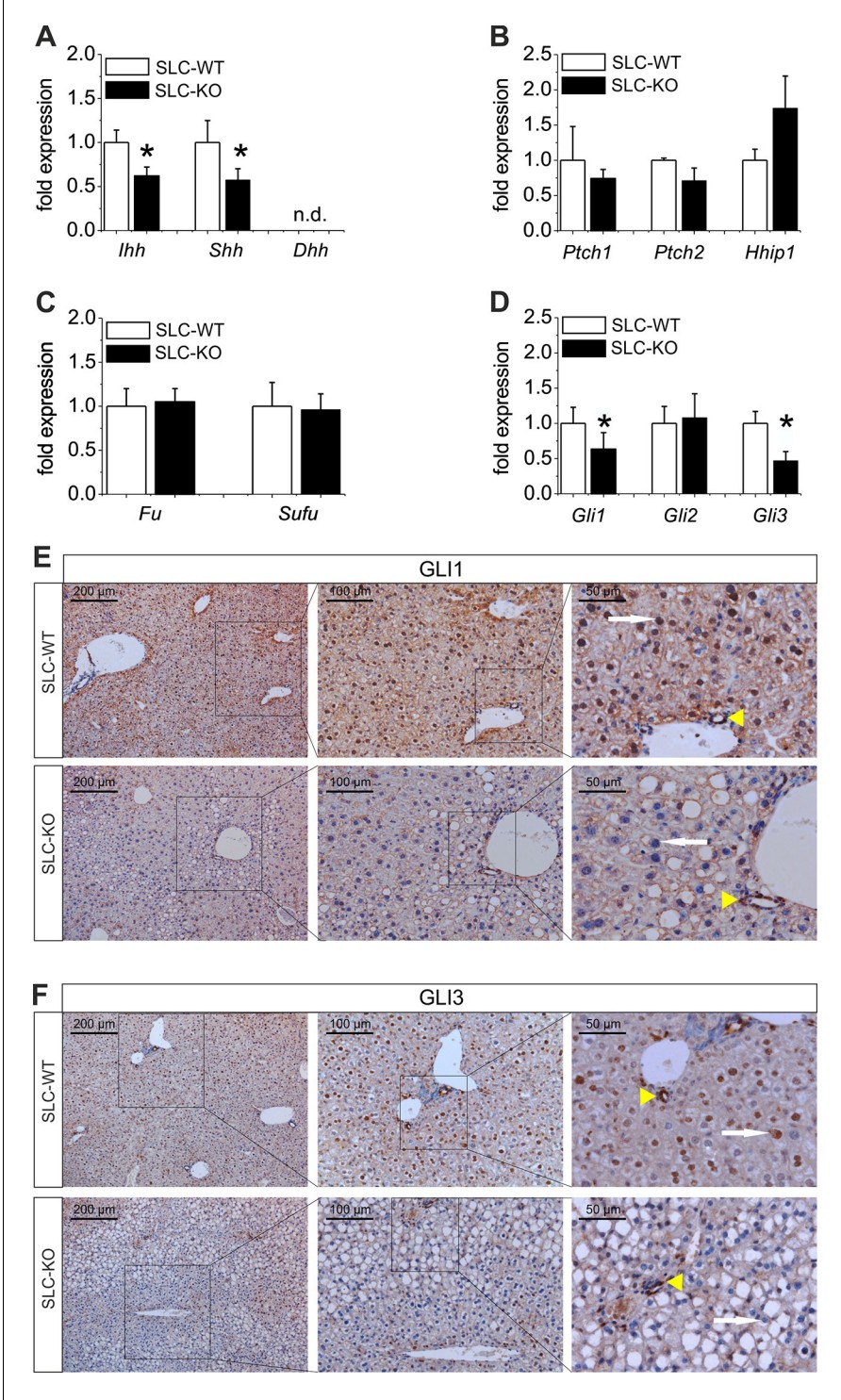

**Figure 3.** Expression of genes and proteins related to Hh signaling in SLC mice. (A–D) qRT-PCR data from isolated hepatocytes from the male SLC-WT (n=6–10) and the SLC-KO (n=6–10) mice illustrating the expression of (A) the ligands Ihh, Shh and Dhh (n.d.: not detectable); (B) the ligand binding molecules *Ptch1*, *Ptch2* and *Hhip1*; (C) the downstream genes *Fu* and *Sufu* and (D) the TFs *Gli1, Gli2* and *Gli3* of the Hh signaling pathway. (E–F) Immunohistochemistry of liver sections from male SLC-WT and SLC-KO mice of (E) GLI1 and (F) GLI3. Labeled hepatocyte nuclei for both Gli factors are seen only in WT, but not in KO animals (white arrows). Staining in non-parenchymal cells, e. g. bile ductular cells (yellow arrowheads) is not affected by the knockout. Scale bars: 200 μm;
*Figure 3 continued on next page*

*Figure 3 continued*

100 μm and 50 μm. Source files of all data used for the quantitative analysis are available in the *Figure 3—source data 1*.

The following source data and figure supplements are available for figure 3:

**Source data 1.** Source data of the expression of genes related to Hh signaling in SLC mice (*Figure 3A–D*).

**Figure supplement 1.** Western Blot of GLI3 in the SLC mice.

**Figure supplement 2.** The "steatotic Gli-code" extents to several mouse models and humans with steatosis.

## *Smo* deficiency up-regulates key lipogenic transcription factors and enzymes

QRT-PCR was used to confirm major results of the microarray analysis and to detect additional regulated genes with higher sensitivity and accuracy. As shown in *Figure 4B*, the significant up-regulation of several TFs involved in the regulation of lipid and carbohydrate metabolism was detected including *Chrebp* (Carbohydrate-responsive element-binding protein), *Srebf1*, *Srebf2* (Sterol regulatory element binding transcription factor 1/2), *Ppara* and *Pparg* (Peroxisome proliferator activated receptor alpha/gamma) in SLC-KO mice. To confirm this data on protein level, immunohistochemical staining was performed for the expression of SREBP1 and PPARG in liver sections of SLC-WT and SLC-KO mice. The SREBP1 protein showed a slightly pericentral and rare nuclear preference in liver parenchyma from SLC-WT mice and was very strongly enhanced in the pericentral zone of livers from SLC-KO mice. In particular, the frequency of nuclear staining in hepatocytes was much higher in these mice compared to control mice (*Figure 4C*).

Likewise, PPARG protein was much more frequent in hepatocyte nuclei of SLC-KO mice than in SLC-WT mice. Also with PPARG heterogeneous distribution of the protein was obvious (*Figure 4D*).

Additionally, the gene expression of the anti-adipogenic TF *Gata4* (*Patankar et al., 2012*) was significantly reduced, whereas *Gata6* remained unchanged (*Figure 5A*). The *Nfyb* (Nuclear transcription factor-Y beta) mRNA was also markedly reduced, while the *Nfyg* (Nuclear transcription factor-Y gamma) and *Lxra* (Liver X receptor alpha) mRNAs remained unaffected. The expression of *Foxa1* and *Foxa2* (Forkhead box A1/2), which are known to mediate the effects of insulin on lipid metabolism (*Wolfrum et al., 2004*), did not change (*Figure 5B*). Interestingly, the expression of *Nr1d2* (Rev-ErbA beta) was strongly down-regulated, while that of *Nr1d1* (Rev-ErbA alpha) remained unchanged (*Figure 5B*).

Of the key lipogenic enzymes, we found increased expression of those involved in the biosynthesis of fatty acids and triglycerides, including *Acaca* (Acetyl-Coenzyme A carboxylase alpha), *Fasn* (fatty acid synthase), *Elovl6* (Elongation of long chain fatty acids) and *Gpam* (Glycerol-3-phosphate acyltransferase) (*Figure 5C*). In contrast, the expression of *Acacb* was not changed, and *Elovl3* expression was significantly decreased (*Figure 5C*). The transcripts for enzymes involved in cholesterol biosynthesis, such as *Aacs* (acetoacetyl-CoA synthetase), *Hmgcr* (3-hydroxy-3-methylglutaryl-Coenzyme A reductase), and *Lss* (Lanosterol synthase), were also increased (*Figure 5D*). These findings suggest a coordinated response of genes favoring fatty acid, triglyceride and cholesterol biosynthesis. Intriguingly, the newly discovered NAFLD-associated gene *Pnpla3* (Adiponutrin) (*Chow et al., 2014*; *Smagris et al., 2015*) was also dramatically increased (*Figure 5D*).

Using this data, the results of our microarray analysis, and published databases, we created a hypothetical protein-protein interaction network (PPI) of the studied steatosis- and Hh-signaling-related genes with the STRINGv10 software (*Figure 5E*). This network provides an overview of the studied proteins and confirms the observed diverse and highly complex connections of the TFs and enzymes related to lipid metabolism and Hh signaling. PNPLA3 expression is connected to SREBF1 activity, and the PPAR family is linked to the FOXA and GLI TFs. The lipogenic enzymes (e.g. FASN, ACACA/B, ELOVL6, and GPAM) are closely connected to the TF SREBF1 and to each other as part of the fatty acid and triglyceride biosynthetic pathways.

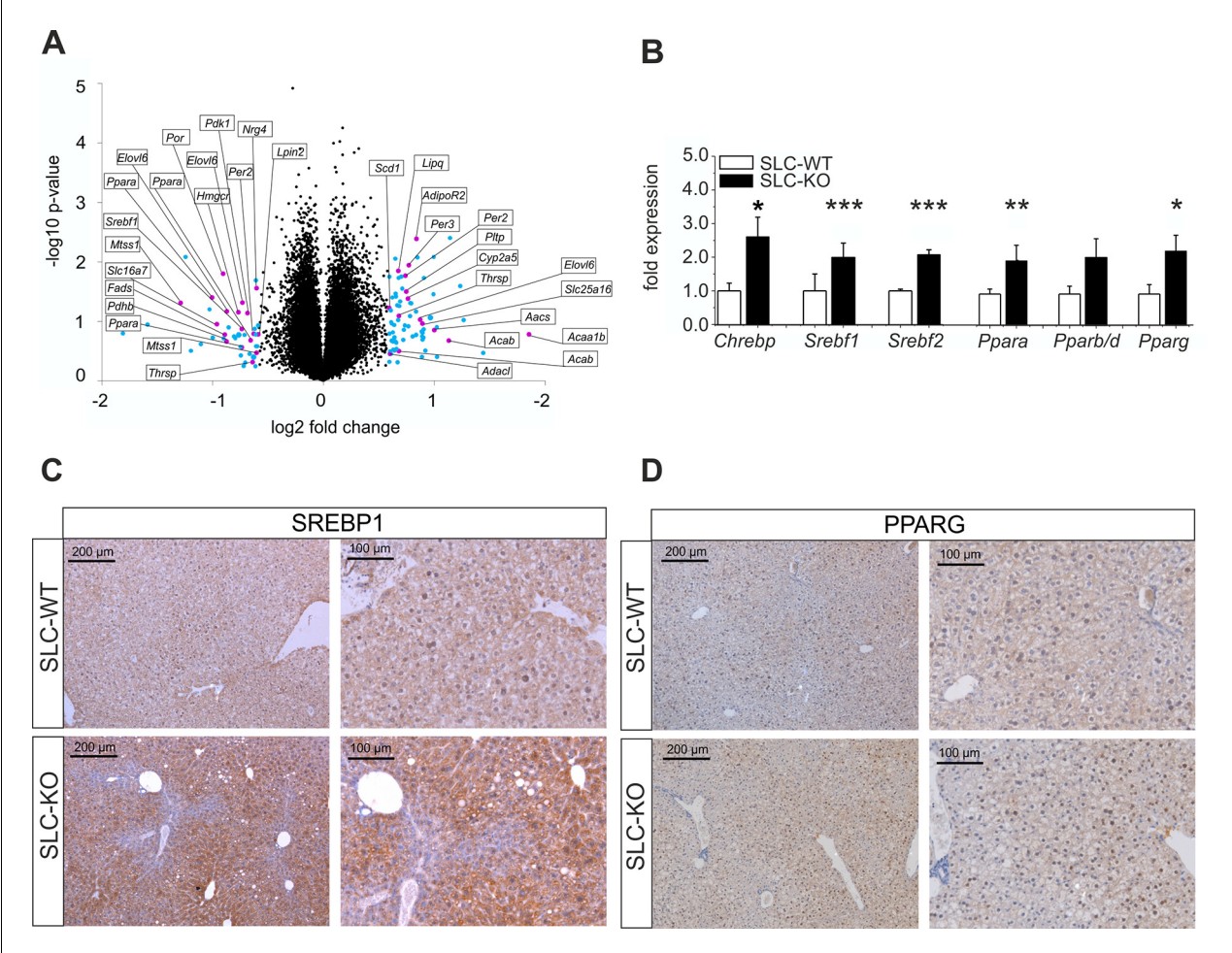

**Figure 4.** Gen and protein expression of hepatic TFs involved in lipid metabolism in SLC mice. (A) Volcano blot visualizing differentially expressed genes in male SLC-KO mice detected by Affymetrix microarray analysis (n=4). All colored dots (blue and magenta) indicate an expression fold change equal or higher than 1.5; magenta: central genes of lipid metabolism; blue: other regulated genes. (B) qRT-PCR of *Chrebp, Srebf1, Srebf2, Ppara, Pparb/d* and *Pparg* from hepatocytes of male SLC-WT (n=6–13) and SLC-KO (n=6–13) mice. (C–D) Immunohistochemistry in liver sections from male SLC-WT and SLC-KO mice. (C) SREBP1 is strongly induced and shows a higher incidence of nuclear staining in pericentral hepatocytes of SLC-KO mice. (D) PPARG shows a much higher incidence in hepatocyte nuclei and a slight cytoplasmic increase in pericentral hepatocytes of SLC-KO mice. Scale bars: 200 µm and 100 µm. Source files of all data used for the quantitative analysis are available in the *Figure 4—source data 2*.

The following source data is available for figure 4:

**Source data 1.** Gene set enrichment analysis of isolated hepatocytes from SLC-WT and SLC-KO mice.

**Source data 2.** Source data of gene expression of hepatic TFs involved in lipid metabolism in SLC mice (*Figure 4B*).

## Coordinated response of enzymes and metabolic pathways

Immunohistochemical analyses of FASN were performed to confirm the differences in expression at the protein level. The stronger staining not only confirmed the up-regulation of FASN in the SLC-KO mice, but, surprisingly, revealed a shift from the known pericentral (*Gebhardt, 1992*; *Postic and Girard, 2008*) to the periportal localization in the SLC-WT mice, matching the preferential site of lipid accumulation (*Figure 6A*). These results suggest that in addition to Wnt/beta-catenin signaling (*Benhamouche et al., 2006*; *Gebhardt and Hovhannisyan, 2010*), Hh signaling is required to maintain the normal zonation of the liver parenchyma, at least with respect to lipid metabolism. Furthermore, we measured the activity of several lipid metabolism pathways in cultured hepatocytes. Fatty acid biosynthesis, as determined by the incorporation of [14C]-labelled acetate, was almost doubled

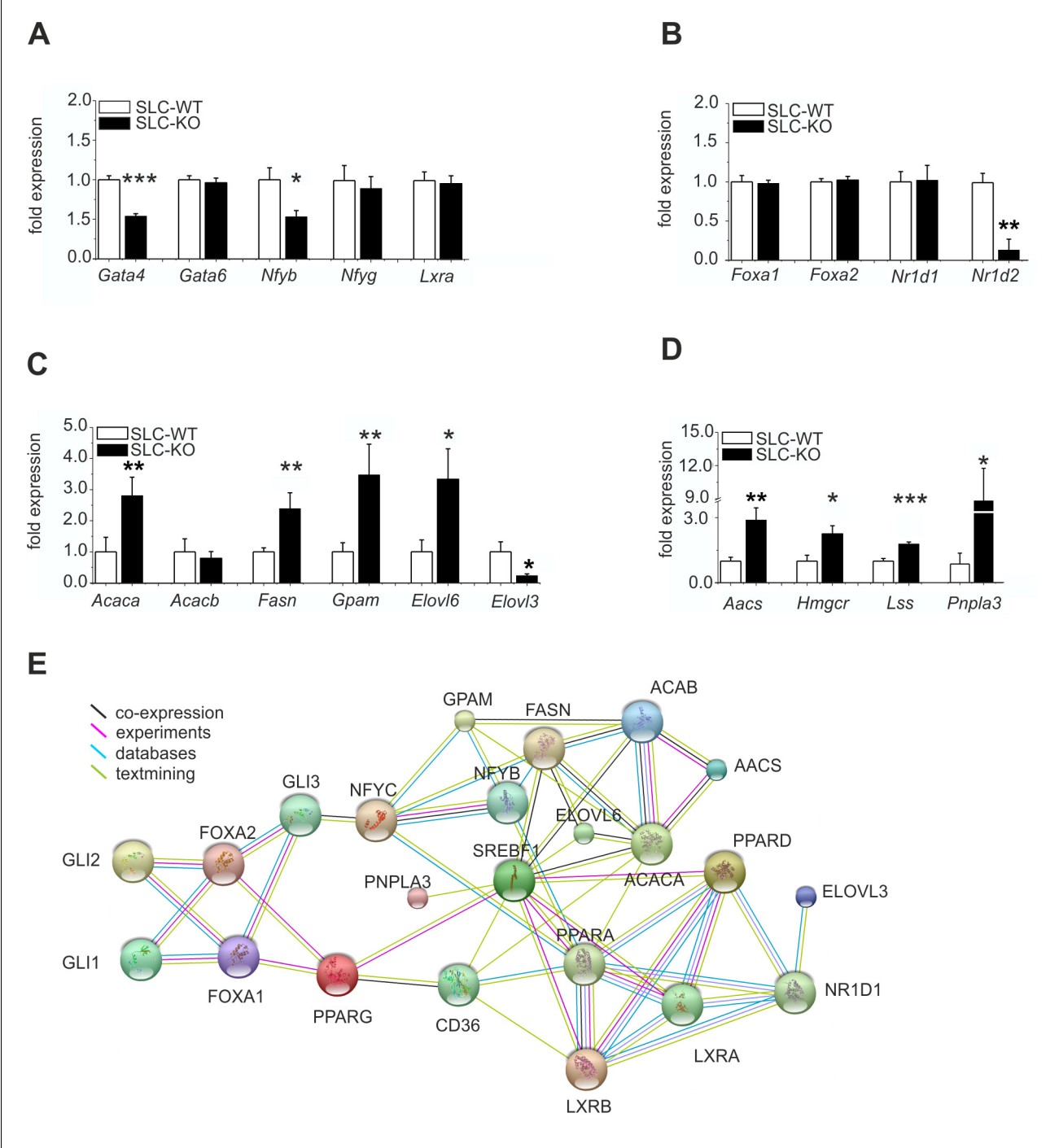

**Figure 5.** Expression of the hepatic TFs and enzymes involved in lipid metabolism in SLC mice. (A–D) qRT-PCR data from hepatocytes from the male SLC-WT (n=6–13) and SLC-KO (n=6–13) mice. (**A**) *Gata4, Gata6, Nfyb, Nfyg* and *Lxra*. (**B**) *Foxa1, Foxa2, Nr1d1* and *Nr1d2*. (**C**) *Acaca, Acacb, Fasn, Gpam, Elovl6* and *Elovl3*. (**D**) *Aacs, Hmgcr, Lss* and *Pnpla3*. (**E**) The PPI (protein-protein interaction) network obtained from the STRINGv10 database using steatosis- and Hh-signaling-related genes as the query. The colored lines indicate co-expression (black), experimental data (purple), database scan (blue) and published scientific abstracts (green). Source files of all data used for the quantitative analysis are available in the *Figure 5—source data 1*.

The following source data is available for figure 5:

**Source data 1.** Source data of expression of the hepatic TFs and enzymes involved in lipid metabolism in SLC mice (*Figure 5A–D*).

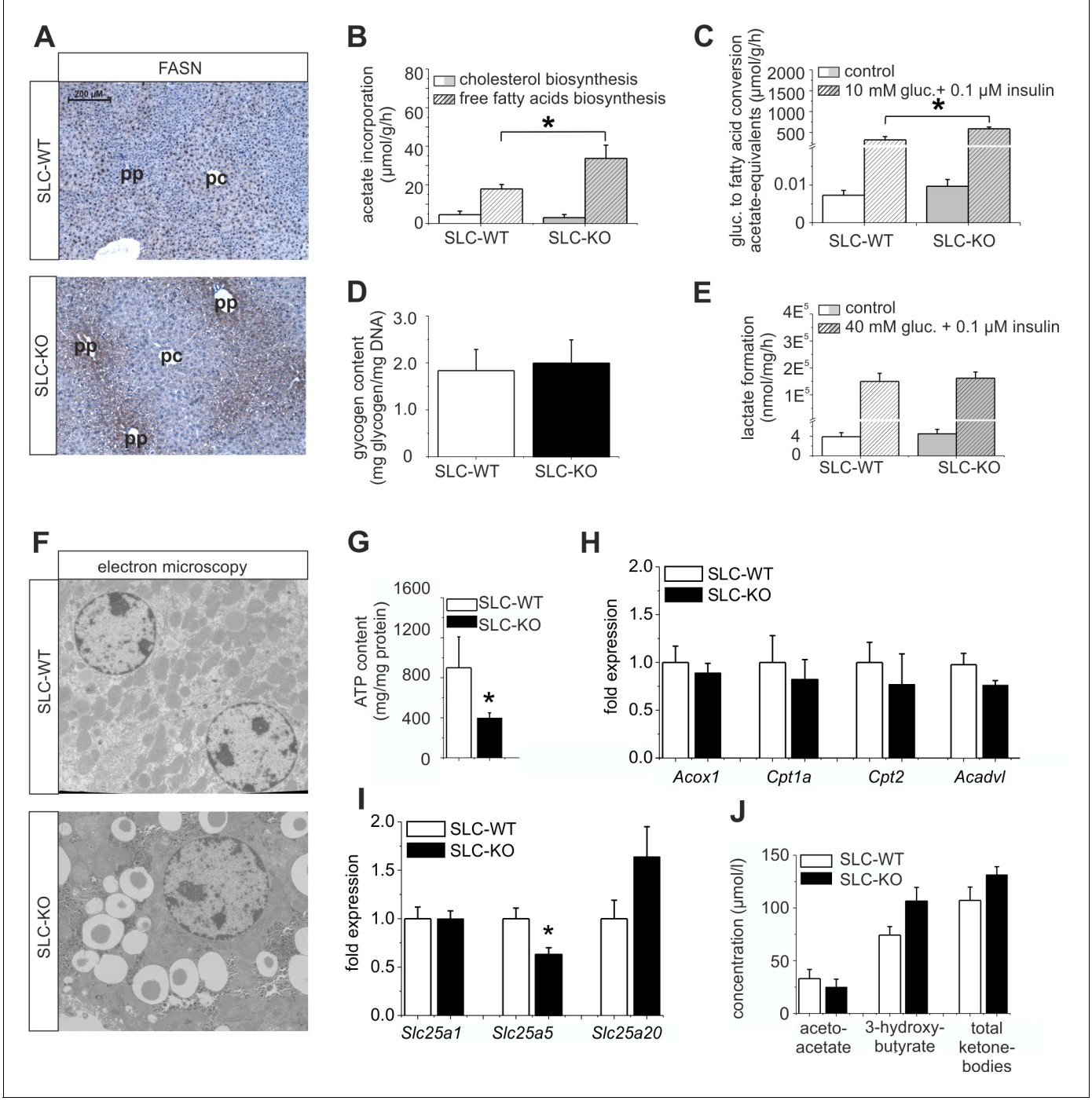

**Figure 6.** Expression of genes and proteins involved in lipid and mitochondrial energy metabolism in SLC mice. (**A**) Immunohistochemistry of FASN in liver sections of male SLC-WT and SLC-KO mice (pp: periportal, pc: pericentral). (**B–H**) Measurements in freshly isolated hepatocytes from male SLC-WT and SLC-KO mice (**B**) Determination of fatty acid and cholesterol biosynthesis. (**C**) Conversion of [$^{14}$C(U)]-glucose to fatty acids. (**D**) Glycogen content. (**E**) Determination of glycolysis. (**F**) Electron microscopy of liver tissue. (**G**) ATP content. (**H–I**) qRT-PCR data from the male SLC-WT (n=6) and the SLC-KO (n=6) mice: (**H**) *Acox1, Cpt1a, Cpt2* and *Acadvl*, (**I**) *Slc25a1, Slc25a2* and *Slc25a20*. (**J**) Serum concentrations of ketone bodies. Eight to ten SLC mice were used in each of the experiments depicted in (**B, C, D, E, G, H, I ,J**). Source files of all data used for the quantitative analysis are available in the *Figure 6—source data 1*.

The following source data and figure supplement are available for figure 6:

**Source data 1.** Source data of expression of genes and proteins involved in lipid and mitochondrial energy metabolism in SLC mice (*Figure 6B–J*).

*Figure 6 continued on next page*

*Figure 6 continued*

**Figure supplement 1.** Expression of respiratory chain complexes in SLC mice.

in the hepatocytes from the SLC-KO mice (*Figure 6B*). However, cholesterol biosynthesis from labelled acetate was not changed in vitro (*Figure 6B*), corresponding to the unchanged serum cholesterol levels *in vivo* (*Figure 1—figure supplement 1A*). Interestingly, the hepatocytes from the SLC-KO mice showed a significantly higher rate of fatty acid synthesis from [$^{14}$C(U)]-glucose in the presence of 10 mM glucose and 0.1 μM insulin (*Figure 6C*), suggesting increased channeling of the high concentrations of glucose into fatty acid biosynthesis. The fact that neither the glycogen content (*Figure 6D*), nor basal or stimulated glycolysis as determined by the conversion of [$^{14}$C(U)]-glucose to lactate (*Figure 6E*) were altered in the hepatocytes from the re-fed SLC-KO mice, suggests that there is no shortage of potential fuel for glycolytic acetyl-CoA formation.

## Changes in mitochondrial functions and associated ATP production

There is growing evidence that mitochondrial dysfunction plays a central role in the pathogenesis of NAFLD (*Petrosillo et al., 2007*; *Wei et al., 2008*). Therefore, we investigated the impact of the Hh pathway on the mitochondria in hepatocytes. Electron microscopy revealed larger and swollen mitochondria in the SLC-KO mice (*Figure 6F*). Further analyses showed significantly reduced ATP levels (*Figure 6G*). However, the activity of the respiratory complexes was not changed (*Figure 6—figure supplement 1*). The expression of carrier proteins (e.g. *Cpt1a, Cpt2*; Carnitine palmitoyltransferases 1a/2) and enzymes (*Acox1*, acyl-Coenzyme A oxidase 1, and *Acadvl*, acyl-Coenzyme A dehydrogenase, very long chain) of beta oxidation were also not changed (*Figure 6H*). Instead, we found the significant decrease of *Slc25a5* (mitochondrial carrier) expression, which transports ADP and ATP through the mitochondrial inner membrane (*Figure 6I*). The fact that no significant changes in the ketone body concentrations could be measured (*Figure 6J*), lead us to speculate that the decrease of *SLC25a5* could be one explanation of the low ATP production.

## Down-regulation of *Gli3* is sufficient for the development of steatosis

Furthermore, we wanted to know whether the down-regulation of *Gli1* and *Gli3* might be sufficient to explain the changes in expression of the TFs and their metabolic consequences; thus, we performed a siRNA-mediated knockdown of the *Gli* factors. *Gli3* knockdown led to a significant increase in the expression of lipogenic TFs including *Ppara, Pparg*, and *Srebf1* (*Figure 7A*). Likewise, only the hepatocytes treated with *Gli3* siRNA responded with an increase in the expression of *Fasn* and *Elovl6* and a reduced expression of *Elovl3* (*Figure 7—figure supplement 1*). To validate the transcriptional data we focused on the lipid content of siRNA-treated cultures. Mainly the hepatocytes transfected with the *Gli3* siRNA showed pronounced accumulation of fat droplets (*Figure 7B*) and elevated lipid staining after 72 hr (*Figure 7C*). To a minor extent *Gli1* knockdown may aid in inducing steatosis, because *Srebf1* was also induced under this condition (*Figure 7A*). *Chrebp* expression did not change in these experiments (*Figure 7—figure supplement 1*), suggesting that the up-regulation observed in the SLC-KO mice (*Figure 4B*) is either due to a GLI-independent mechanism or reflects another adaptive response in vivo.

In order to evaluate the role of GLI3 in mediating the influence of the Hh pathway activity on the expression of downstream lipogenic TFs (*Ppara* and *Srebf1*) and enzymes (*Fasn*) several chromatin immunoprecipitation experiments (ChIP) were performed. Using data mining by MotivMap analyses (*Daily et al., 2011*) several possible binding sites for the GLI3 protein within the range of − 10000 bp upstream to 10000 bp downstream of the transcription start site of the *Fasn, Ppara* and *Srebf1* genes were identified. As consensus sequence we selected TGTGTGGTC for the ChIP analysis (*Figure 7—figure supplement 2A*). The data provided in *Figure 7—figure supplement 2B* give a first hint that direct binding of GLI3 to the predicted binding site in the promoter region of *Srebf1* (at -1156 bp) might be involved in transmitting GLI factor activity to the expression of SREBP1. No such hint was found, however, in the case of *Fasn* (at the predicted binding site at -1767 bp) and *Ppara*

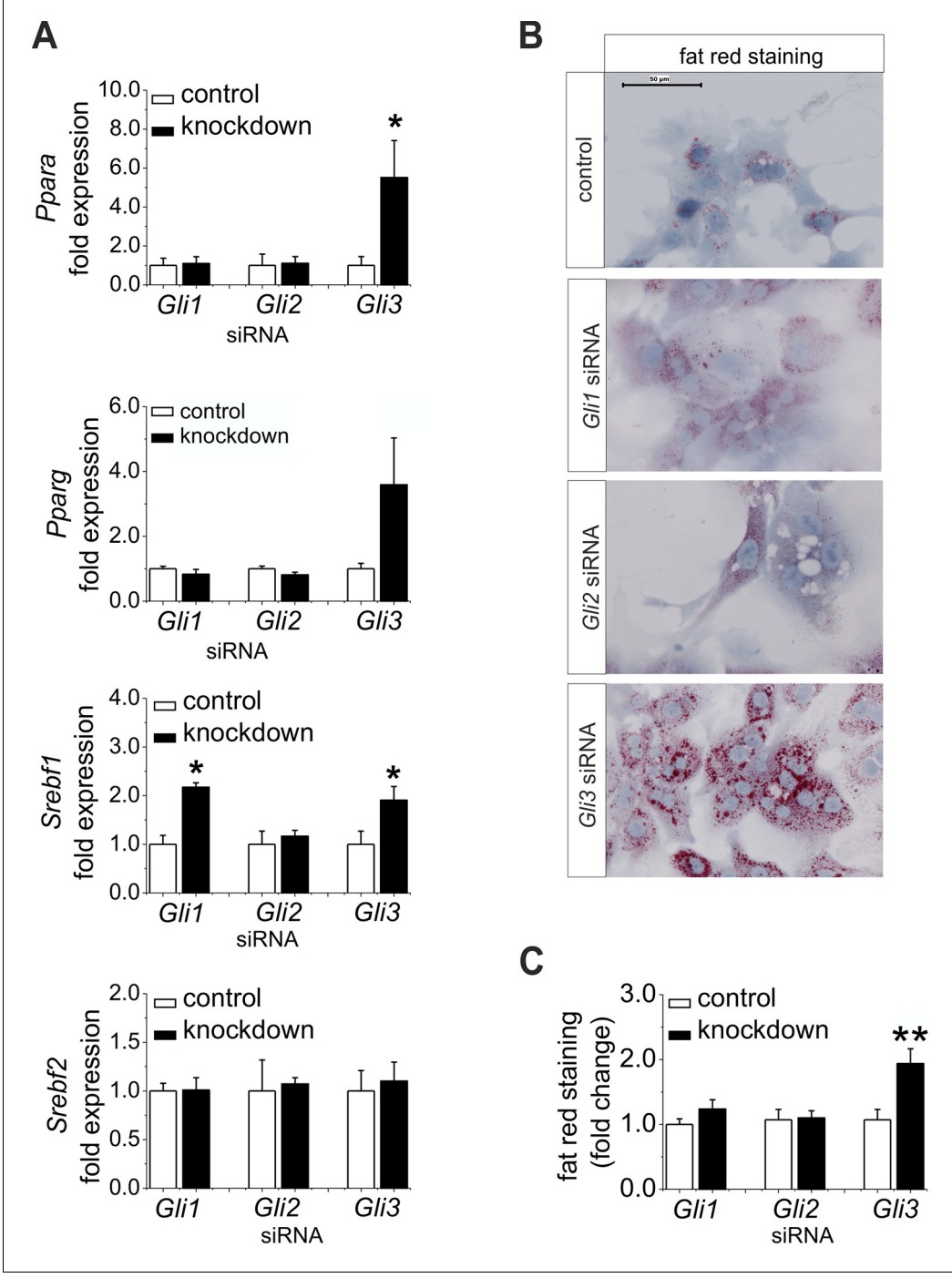

**Figure 7.** Influence of siRNA-mediated knockdown of *Gli1*, *Gli2* and *Gli3* on expression of genes of lipid metabolism. (A–C) Isolated hepatocytes from male C57BL6/N mice (n=4–9) treated with the control, *Gli1*, *Gli2* and *Gli3* siRNAs. (A) Relative expression of *Ppara*, *Pparg*, *Srebf1* and *Srebf2*, determined by qRT-PCR. Data for *Ppara*, *Pparg*, *Srebf1* and *Srebf2* were taken from our publication (*Schmidt-Heck et al., 2015*) with the same mouse model for simplifying comparison. (B) Qualitative and (C) quantitative fat red staining. Source files of all data used for the quantitative analysis are available in the *Figure 7—source data 1*.

The following source data and figure supplements are available for figure 7:

**Source data 1.** Source data of the influence of siRNA-mediated knockdown of *Gli1*, *Gli2* and *Gli3* on expression of genes of lipid metabolism(*Figure 7A,C*).

*Figure 7 continued on next page*

*Figure 7 continued*

**Figure supplement 1.** Influence of siRNA-mediated knockdown of *Gli1, Gli2* and *Gli3* on gene expression of lipogenic enzymes.

**Figure supplement 2.** Chromatin immunoprecipitation experiments for GLI3 binding sites.

(at the predicted binding site at -2674 bp) making it more likely that these proteins are not primary, but secondary targets of GLI3 (mediated among else by SREBP1, see below).

## Activation of Hh signaling is able to prevent steatosis

Finally, we investigated whether it is possible to prevent steatosis via activation of the Hh pathway. Therefore, we used several activating strategies including (i) the knockdown of the suppressor of fused (*Sufu*) (a well-known negative regulator of the signaling cascade; *Law et al., 2012*), (ii) the incubation with SMO agonists (SAG) or recombinant SHH, and (iii) GLI overexpression in hepatocytes from different mouse models.

Ad (i), *Sufu* siRNA significantly reduced *Sufu* mRNA levels (*Figure 8A*) and strongly up-regulate-dexpression of all *Gli* TFs (*Figure 8B*) in hepatocytes from C57BL6/N mice. Regarding the lipogenic TFs, a significant decrease of *Srebf1* and a slight down-regulation of *Srebf2* and *Elovl6* were detected in the treated cells (*Figure 8C*) indicating that up-regulation of the Gli TFs, i. e. reversal of the steatotic GLI code, causes anti-steatotic modulation of lipogenic genes. To go a step further, we determined whether *Sufu* knockdown could reduce hepatic steatosis. Using isolated hepatocytes from MC4R-KO mice, which retain their steatotic phenotype in vitro, we demonstrated a loss of

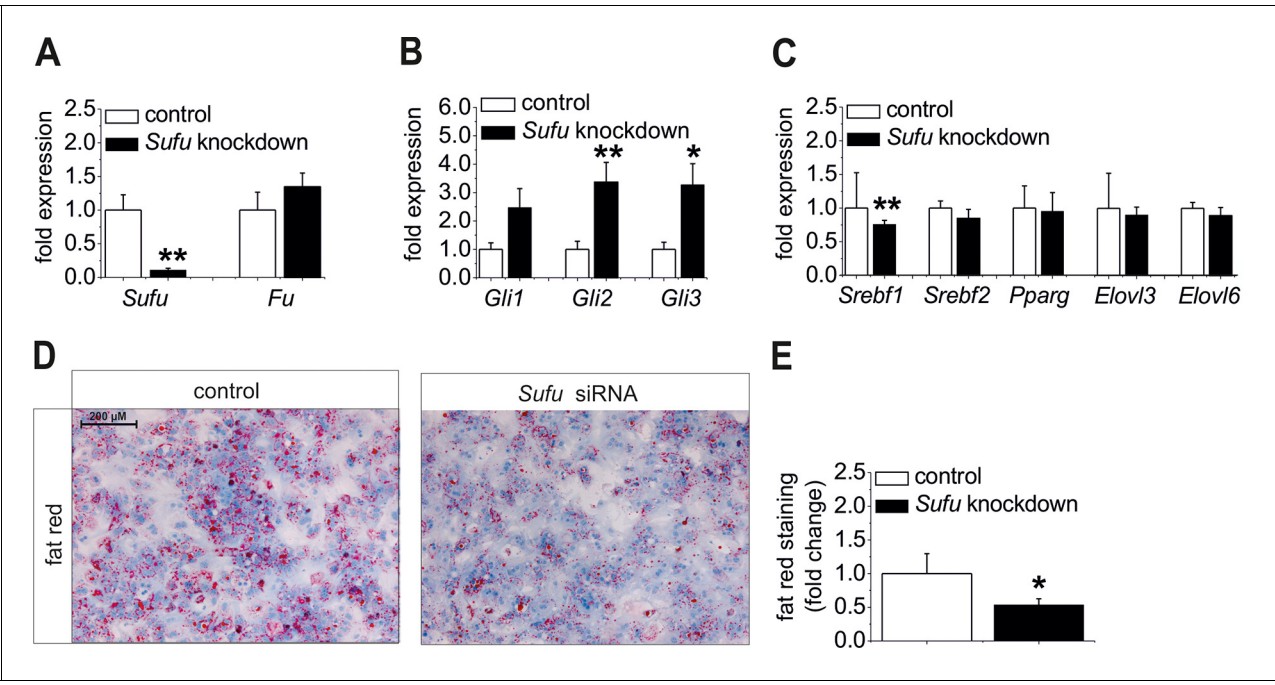

**Figure 8.** Influence of siRNA-mediated knockdown of *Sufu* on lipid metabolism in vitro. (A-C) Isolated hepatocytes from male C57BL6/N mice (n=8) treated with the control and *Sufu* siRNA. Relative expression of the following genes was determined by qRT-PCR 48 hr post-transfection (A) *Sufu* and *Fu*, (B) *Gli1, Gli2* and *Gli3*, (C) *Srebf1, Srebf2, Pparg, Elovl3* and *Elovl6*. (D) Qualitative and (E) quantitative fat red staining in hepatocytes isolated from the MC4R-KO mice in response to treatment with the control and *Sufu* siRNA. Source files of all data used for the quantitative analysis are available in the *Figure 8—source data 1*.

The following source data is available for figure 8:

**Source data 1.** Data source of the influence of siRNA-mediated knockdown of *Sufu* on lipid metabolism in vitro(*Figure 8 A-C, D*).

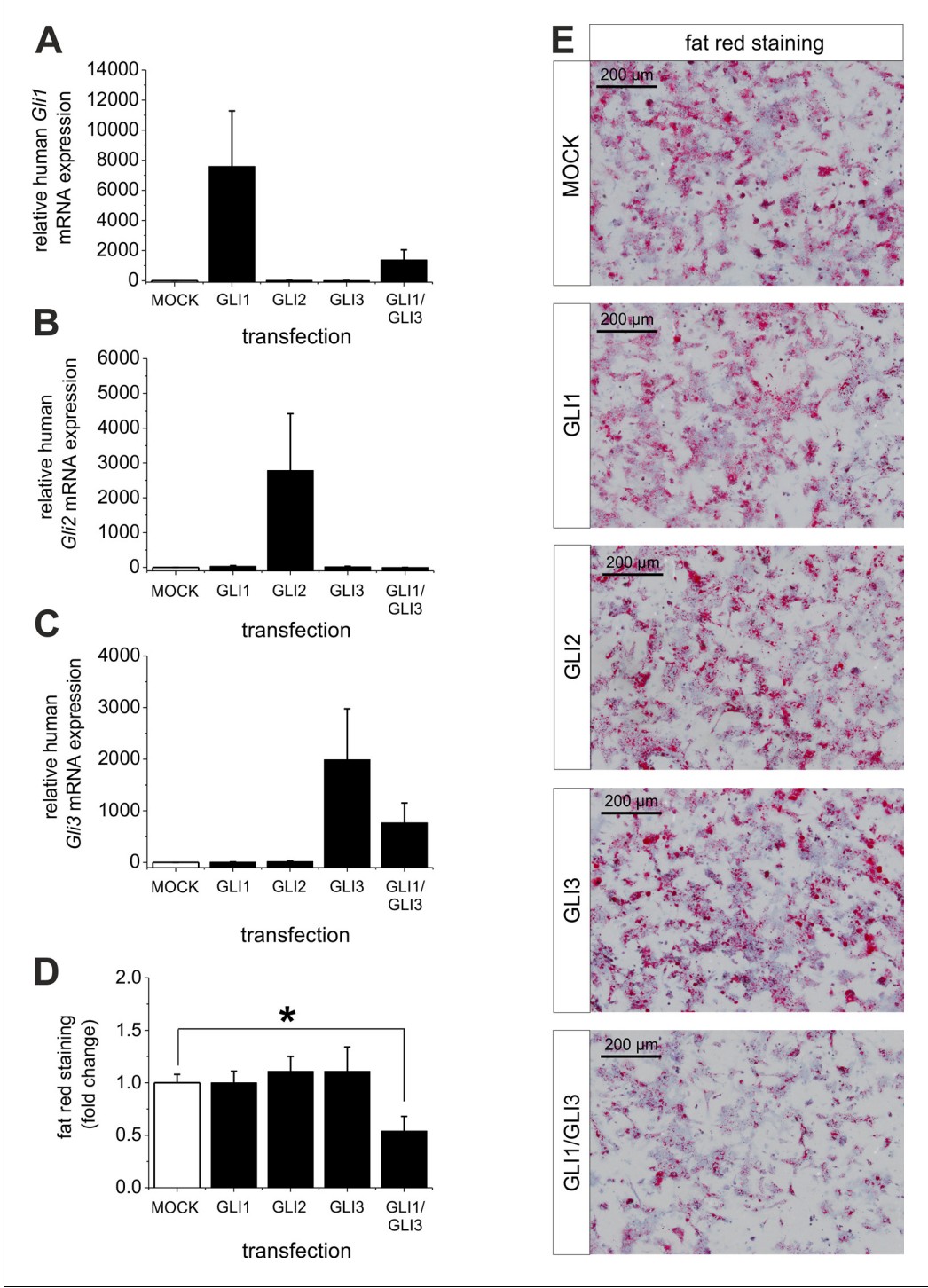

**Figure 9.** Influence of GLI1, GLI2 and GLI3 overexpression on lipid content in hepatocytes of ob/ob mice. Isolated hepatocytes from male ob/ob mice (n=3) transfected with the control plasmid (MOCK) and human overexpression plasmids of GLI1, GLI2, and GLI3 or the combination of GLI1 and GLI3, 72 hr post-transfection as described in Materials and Methods. (A-C) Relative expression of the following genes was determined by qRT-PCR. (A) *Gli1*, (B) *Gli2* (C) *Gli3*. (D) Quantitative and (E) qualitative fat red staining. Lipid content was reduced only in the presence of the combined overexpression of GLI1 and GLI3, but not after expression of each Gli factor alone. Source files of all data used for the quantitative analysis are available in the *Figure 9— source data 1*.

The following source data is available for figure 9:

**Source data 1.** Source data of the influence of GLI1, GLI2 and GLI3 overexpression on lipid content in hepatocytes of ob/ob mice (*Figure A-D*).

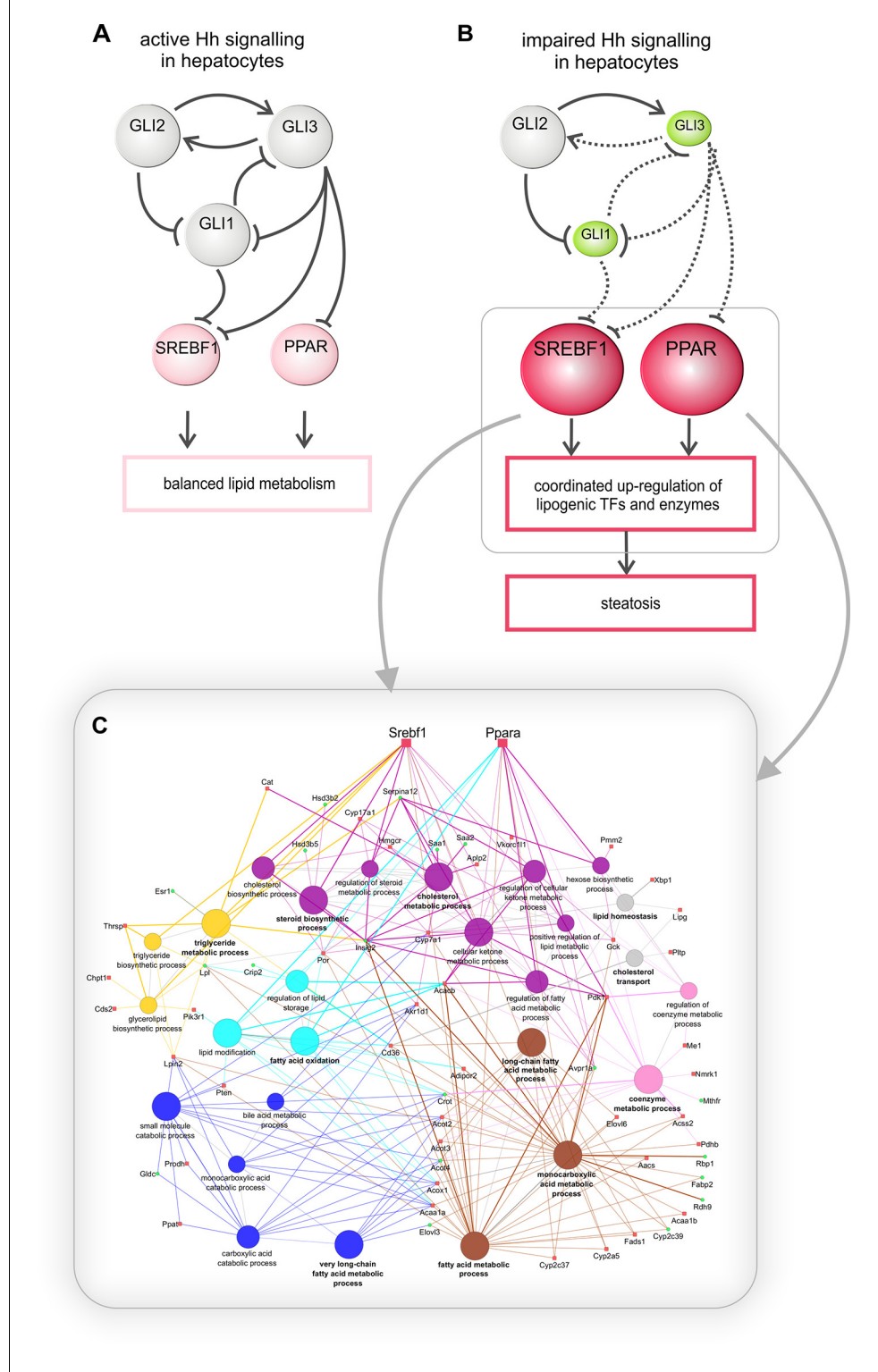

**Figure 10.** Regulation of hepatic lipid metabolism by Hh signaling. (**A**) Intact Hh signaling allows a self-supporting network of the GLI TFs (***Schmidt-Heck et al., 2015***). *Gli1* and *Gli3* exert attenuating effects on lipogenic TFs resulting in balanced liver lipid metabolism. (**B**) The impairment of Hh signaling down-regulates (green) *Gli1* and *Gli3*, while *Gli2* is not affected. These selective changes, in turn, lead to the up-regulated expression (red) of *Srebf1* and members of the *Ppar* family and eventually other TFs. (**C**) These secondary TFs contribute to a complex

*Figure 10 continued on next page*

*Figure 10 continued*

regulatory network leading to the up-regulation of lipogenic enzymes , which ultimately cause steatosis. The dashed lines indicate weaker effects than the solid lines.

staining intensity in the cultures treated with the *Sufu* siRNA compared with the cultures transfected with the control (*Figure 8D,E*).

Ad (ii), We also could observed a significant reduction in fat red staining when the steatotic hepatocytes from the MC4R-KO and *ob/ob* mice were incubated with the SMO agonist SAG and recombinant SHH (*Figure 3—figure supplement 2D, E*). These experiments indicate that activation of Hh signaling upstream of *Sufu* on the Gli TF level by an Hh agonist or an Hh ligand has similar anti-steatotic effects as the *Sufu* knockdown.

Ad (iii), to proof the assumption that reversal of the level of the Gli TFs (GLI1 and GLI3) is able to overcome steatosis we treated isolated hepatocytes from steatotic ob/ob mice with overexpression plasmids for GLI1, GLI2, GLI3 and a combination of GLI1 and GLI3. The results clearly show that the transfection of the GLI plasmids leads to a strong increase in the corresponding *Gli* mRNA expression, while the combined transfection of GLI1 and GLI3 expression plasmids resulted in a slightly lower expression of the corresponding *Gli* mRNAs (*Figure 9A–C*). The results from quantitative and qualitative analysis of fat red staining indicate that overexpression of one Gli factor alone is not able to reduce the lipid content. However, the combination of GLI1 and GLI3 significantly mitigated the steatosis in the hepatocytes from ob/ob mice (*Figure 9D,E*) indicating that parallel changes in both GLI factors are required to reverse steatosis.

Interestingly, quantitative and qualitative determination of fat red staining demonstrates that only the combined overexpression of GLI1 and GLI3 (but not of each Gli factor alone) is able to significantly reduce the lipid content in the hepatocytes from ob/ob mice (*Figure 9D,E*). These results emphasize the strong influence of the GLI-code and, thus, the mediating function of Hh signaling in an animal model of steatosis of completely different origin (i.e leptin deficiency).

## Discussion

In this study, we demonstrate that the hepatocyte-specific impairment of canonical Hh signaling by the conditional ablation of *Smo* results in considerable changes in liver lipid metabolism that ultimately lead to hepatic steatosis. Hence, the undisturbed Hh pathway is necessary to maintain the proper balance between the synthesis/uptake and degradation/export of fatty acids and triglycerides in adult hepatocytes (*Figure 10A*). These results were confirmed on the metabolic and the transcriptional level. Metabolically, we demonstrated the increased accumulation of triglycerides and the significant increase of VLDL particles, which are a hallmark of hepatic steatosis and NAFLD (*Fon Tacer and Rozman, 2011*). On the transcriptional level, we showed that a large set of lipogenic TFs and enzymes were significantly up-regulated following the deletion of *Smo*. Among the responding TFs were *Srebf1*, *Ppara* and *Pparg* strongly increased which are characteristic for liver steatosis (*Fabbrini et al., 2010*; *Ables, 2012*). Interestingly, combinatorial interactions between *Srepf1*, *Nfyb* and *Nfyg* (Nuclear transcription factor Y), and *Sp1* (Transacting transcription factor 1) have been shown to determine the expression of specific sets of target genes, including hundreds of genes with distinct roles in lipid metabolism and other functions (*Reed et al., 2008*). Especially the role of SREBP1 is further discussed below. Some studies with certain hepatic cells already revealed the contributions of *Nfyb/g* to the overexpression of lipid metabolizing enzymes and several other metabolic events (*Woo et al., 2005*; *Damiano et al., 2009*). Likewise, the differential response of *Nr1d1* and *Nr1d2*, which relate the circadian rhythm to metabolism and disease (*Duez and Staels, 2008*; *Ramanathan et al., 2014*), may indicate that Hh signaling in hepatocytes could also influence the circadian fluctuations of lipid metabolism.

Further support for a highly complex but coordinated response to down-regulation of Hh pathway activity in hepatocytes that shifts the normal balance of lipid metabolism in favour of liponeogenesis is provided by the observed response of the genes encoding the central lipogenic enzymes. For instance, increased *Acaca* but not *Acacb* expression (*Chow et al., 2014*) and increased *Elovl6* expression, particularly when coupled to the down-regulation of *Elovl3*, should trigger lipogenesis

(*Jakobsson et al., 2005*). The genome-wide dimension of the role of Hh signaling in controlling central transcriptional events of NAFLD formation and progression was revealed by our microarray studies. In particular, the GSEA revealed that many regulated genes in SLC-KO mice belong to the top 50 abundant changed genes associated with NAFLD tested in 100 unique inbred mouse strains (*Hui et al., 2015*). Intriguingly, one of these changes concerned the increased expression of *Pnpla3* which has frequently been found to be associated with steatosis where it acts as a downstream gene of SREBP-1c to promote lipid accumulation (*Smagris et al., 2015*). Our observation that liponeogenesis is further increased in the presence of high glucose concentrations adds another piece of the puzzle. Thus, Hh signaling may also influence carbohydrate metabolism in the presence of excess glucose, possibly by increasing *Chrebp* expression.

Another important result of our investigation is that the Hh pathway appears to play an important role in liver zonation, which is shown by the completely changed zonation of FASN in response to the loss of *Smo*. This result points to a deeper impact of this pathway not only on lipid metabolism but also on Metabolic Zonation as a whole (*Schleicher et al., 2015*). Most importantly, the strong enhancement of SREBP1 und PPRAG protein expression in the SLC-KO mice occurred in the pericentral zone which is in line with published data from mice fed with a high fat diet (*Inoue et al., 2005*; *Liu, 2012*). For SREBP1 it was shown that this protein has binding sites at -7000 bp and -500 bp of the *Fasn* gene (*Amemiya-Kudo et al., 2002*; *Morishita et al., 2014*). Furthermore it was also shown that overexpression of SREBP1c, an isoform of SREBP1, lead to significant upregulation of *Fasn* gene expression in hepatocytes (*Dentin et al., 2004*). Obviously, SREBP1 does not act alone but in combination with several other TFs such as the NFYs and SP1 (*Reed et al., 2008*).

To identify the downstream signaling pathways that lead to the observed lipid accumulations in hepatocytes, we extended our previous work on the Gli TFs, which form a self-stabilizing network in parenchymal cells (*Schmidt-Heck et al., 2015*). This network mediates the prominent up-regulation of lipogenic TFs (e.g. *Ppara*, *Pparg* and *Srebf1*) when hepatocytes are treated with *Gli3* siRNA (*Schmidt-Heck et al., 2015*). In our current study, we revealed additional *Gli*-dependent transcriptional effects. Even more importantly, we found that the expression of *Gli1* and *Gli3* were significantly reduced in the SLC-KO mice. This indicates that a specific transcriptional signature within the *Gli* TF network which can be termed the 'steatotic Gli-code' lead to the observed steatosis *in vivo* and *in vitro*.This signature is characterized by reduced expression of *Gli1* and *Gli3*, while *Gli2* is unchanged (or slightly increased), which can be abbreviated as $_{Gli1}$/Gli2/$_{Gli3}$. Using different mouse models with strong steatosis by hyperphagy or leptin deficiency (e.g. MC4R and *ob/ob* mice), we observed the same direction of the regulation of the *Gli* TF's. This was also supported by measurements of *Gli* expression in liver samples from human patients with steatosis. These results indicate that the postulated "steatotic Gli-code" is universal across several strains and species promoting lipid accumulation in hepatocytes regardless of the manner of the etiology of steatosis. Based on the different functional possibilities of the GLI proteins, Ruiz I Altaba has originally postulated the concept of the 'Gli-code', which describes the combinatorial and cooperative function of the GLI transcription factors (*Ruiz i Altaba, 1999*; *Ruiz i Altaba et al.,* 2003; *2007*). This concept has proven useful in connecting different levels of Hh signaling with the regulation of cell fates and cancer (*Stecca and Ruiz i Altaba, 2010*). Because the GLI proteins are also regulated by other signaling pathways, e.g. EGFR-MEK/ERK, RAS/AKT, TGF-beta and Wnt/beta-catenin signaling (for review see [*Fernandez-Zapico, 2008*]), the 'Gli-code' may reflect the outcome of integrating different signaling pathways. Regarding lipid metabolic processes, Suh and co-workers found similar changes in the expression of the *Gli* factors during adipogenic induction of 3T3-L1 fibroblasts (*Suh et al., 2006*). These similarities lead us to hypothesize that the transition from the normal GLI-code (GLI1/GLI2/GLI3) to the postulated 'steatotic Gli-code' ($_{GLI1}$/GLI2/$_{GLI3}$) provides a novel paradigm for the regulation of lipid metabolism in the liver (*Figure 10B*). In case of down-regulated *Gli1* and *Gli3,* prominent lipogenic transcription factors (e.g. *Ppara/g*, *Srebf1*) are up-regulated and influence an entire network of several genes associated with lipid metabolism which finally ends up with hepatic steatosis (*Figure 10B*). This concept is considerably supported by the finding that the combined overexpression of GLI1 and GLI3 (but not of each GLI factor alone) is able to significantly lower the lipid content of hepatocytes from steatotic ob/ob mice.

In order to illustrate the complexity of the downstream gene sets affected by the GLI factor network the 55 overrepresented GO categories identified in the ClueGO analysis were reduced to 13 groups by grouping similar GO categories. *Figure 10C* shows the 7 selected groups (out of the 13)

that are related to lipid metabolism only. Besides direct GLI-dependent regulation also possible indirect mechanisms may be included.

Using ChIP experiments we could provide first hints for a direct regulation of *Srebf1* gene expression via binding sites for the GLI3 protein in its promotor region. However, given the complex interactions of the GLI factors with regulatory binding sites of target genes and the lack of reliable antibodies for full length, truncated and/or phosphorylated GLI factors the results of the CHIP analysis deserve confirmation by detailed analyses using different molecular techniques. Importantly, work by Gurdziel and coworkers (*Gurdziel et al., 2016*) published during revision of our manuscript supports our findings by identifying a comprehensive library of enriched GLI binding motifs in which among else the promotor regions of *Srebf1* and *Pnpla3* are predicted to harbor Hh activity via GLI binding sites (*Gurdziel et al., 2016*).

Even the down-regulation of IGF1 which we observed in the SLC-KO mice and the closely related SAC mice (*Matz-Soja et al., 2014*) may cause indirect alterations in the transcriptome in the liver and in other organs. Whether these effects lead to hepatic changes in the expression of *Chrebp* which does not seem to be a direct target of the GLI factors remains to be established.

The relevance of our findings for the human liver is emphasized by our observation that increased lipid droplets occur in human hepatocytes when the Hh pathway was inhibited by cyclopamine. This fits to similar results reported for HepG2 cells . Furthermore, the obvious down-regulation of *Gli1* and *Gli3* in human livers with steatosis supports the universality of the postulated 'steatotic Gli-code' and reveals new insights in the regulation of hepatic lipid metabolism. In addition, the strong up-regulation of *Pnpla3* also correlates with the expression changes observed in human NAFLD (*Romeo et al., 2008*). Finally, we would like to point out that the already published changes in the IGF-Axis due to impairment of Hedgehog signaling (*Matz-Soja et al., 2014*) are even a hallmark for human steatosis. Both, the dramatic down-regulation of hepatic *Igf1* expression and the upregulation of *Igfbp1*, which were reflected in changes of the serum levels of these proteins, were observed in several clinical studies and are strongly associated with NAFLD and the metabolic syndrome (*Völzke et al., 2009*; *Alderete et al., 2011*; *Mallea-Gil et al., 2012*).

The central role of the Hh signaling pathway in liver steatosis is emphasized by our findings in vitro that activation of Hh signaling via the *Sufu* siRNA, recombinant SHH, the Hh agonist SAG and the combined overexpression of GLI1 and GLI3 was able to significantly reduce the expression of *Srebf1* and mitigate the accumulation of lipids in steatotic hepatocytes. These findings suggest possible new therapeutic strategies for NAFLD. However, a limiting factor may be that high over-activation of Hh signaling may enhance the risk for carcinogenesis .

Collectively, our study reveals an important role for Hh signaling in regulating hepatic lipid metabolism and its zonation. These findings suggest a new paradigm for the development of liver steatosis. The potential of impaired Hh signaling to trigger steatosis independent of nutritional changes suggests that malfunctions in this pathway may pave the way for the development of NAFLD long before other cues may lead to further aggravation.

## Materials and methods

### Sample-size estimation
When the study was being designed the appropriate sample size was computed using the Sigma Plot software with a power of 0.8 and alpha = 0.05. In addition we used also our experience from previous studies dealing with liver related investigations with the transgenic and even the non-transgenic mice (*Matz-Soja et al., 2014*; *Schmidt-Heck et al., 2015*). For isolated hepatocytes from transgenic SLC-WT, SLC-KO and non-transgenic C57BL6/N mice the same procedure was used. For assays with cultured hepatocytes (siRNA experiments, use of agonists and inhibitors) we used the paired t-Test with a power of 0.8 and alpha = 0.05 to calculate the required sample size.

### Replicates, determination of outliers and exclusion/inclusion of samples
Most experiments were performed 2–3 times with different numbers of biological replicates (animals) of n = 4–17 as indicated in each figure. The number of technical replicates depended on the specific type of measurement and was duplicate in most cases and triplicate in some few experiments.

Outliers were analyzed using the ROUT method on GraphPad Prism 6 software. The aggressiveness of the test was set to 0.2 %. The cleaned data was used for the subsequent statistical analyzes and data depiction in the figures.

By using material from transgenic mice with *Smo* deletion, the amount of *Smo* mRNA was quantified via qRT-PCR. When the remaining expression in liver and hepatocytes from SLC-KO mice was more than 50 % of the mean of the SLC-WT mice, we excluded the sample. Only samples where the expression of *Smo* was lower than 50 % were included.

## Maintenance of the mice, feeding and determination of food consumption

The mice were maintained in a pathogen-free facility on a 12:12 hr LD cycle, according to the German guidelines and the world medical association declaration of Helsinki for the care and safe use of experimental animals. The animals had free access to regular chow (sniff M-Z V1124-0 composed of 22.0 % protein, 50.1 % carbohydrate, 4.5 % fat; usable energy: 13.7 kJ/g; ssniff Spezialdiäten GmbH, Germany) and tap water throughout life. Before sacrifice (between 9 and 11 am), the mice were starved for 24 hr and re-fed with regular chow for 12 hr to obtain a synchronized feeding state.

## Isolation and cultivation of primary hepatocytes and human liver

Primary hepatocytes were isolated from male transgenic and C57BL/6N mice using collagenase perfusion of the liver, as previously described (*Gebhardt et al., 2010*; *Matz-Soja et al., 2014*). The cell suspension was cleared of the non-parenchymal cells by differential centrifugation. Finally, the hepatocytes were suspended in Williams Medium E containing 10 % fetal calf serum and the indicated additions and were plated in 6-well or 12-well plates that had been pre-coated with collagen type I (*Klingmuller et al., 2006*). After 4 hr, the cells were switched to serum-free medium, which was used throughout cultivation.

Cryopreserved human hepatocytes were purchased from TebuBio (Germany). They were thawed according to existing protocols (*Klingmuller et al., 2006*) and cultured in 6-well plates at the same cell density and culture conditions as the mouse hepatocytes, except for the omission of dexamethasone after 4 hr. The hepatocytes were incubated in the presence of 10 µM cyclopamine or vehicle (0.1 % DMSO) control for 72 hr. In some experiments, 300 nM SAG (N-Methyl-N'-(3-pyridinylbenzyl)-N'-(3-chlorobenzo[b]thiophene-2-carbonyl)-1,4-diaminocyclohexane) (Sigma-Aldrich, Germany) was used and DMSO was used as the vehicle. In other experiments, 0.25 µg of recombinant SHH (R&D Systems, USA) or an equivalent amount of vehicle (PBS, containing 0.1 % bovine serum albumin) were used.

Human liver tissues for mRNA expression analysis were obtained from patients without and patients with simple steatosis. Steatosis was histologically examined by a pathologist. Surgery was done because of hepatic metastases of extrahepatic tumors and only healthy (non-tumorous) tissue was used. Experimental procedures were performed according to the guidelines of the charitable state controlled foundation Human Tissue and Cell Research (HTCR), with the written informed patient's consent approved by the local ethical committee of the University of Regensburg (12-101-0048).

## Serum parameters

Blood samples were taken from the beating heart of anesthetized mice. Insulin was detected in the serum using the ELISA kits from DRG Instruments (EIA 3439; Mediagnost, Germany). The serum enzyme activities of ASAT (aspartate aminotransferase), ALAT (alanine aminotransferase) and GLDH (glutamate dehydrogenase) and the serum concentrations of ketone bodies were measured with an automated analyzer (Roche modular) using standardized assays (Roche, Germany). Lipoproteins were isolated by sequential ultracentrifugation from 60 µl of plasma at densities (d) of <1.006 g/ml (very low density lipoprotein, VLDL), d≤1.063 g/ml (intermediate density lipoprotein and low density lipoprotein, LDL) and d>1.063 g/ml (high density lipoprotein, HDL) in a LE-80K ultracentrifuge (Beckman, Germany) as described. Cholesterol in the lipoprotein fractions was determined enzymatically using a colorimetric method (Roche, Germany). The blood glucose levels were determined using a blood glucose meter (Freestyle Mini, Abbott, Germany).

## Histological analysis

Paraffin sections (3 μm) were stained with H&E to visualize the tissue histology. For electron microscopy, separate small pieces of liver tissue (approx. 1 mm$^3$) were fixed in 2.5% glutaraldehyde in PBS (pH 7.6). Further processing was performed as previously described (*Gebhardt, 1992*). Frozen sections were cut at 6 μm for the quantitative and qualitative lipid analysis. A Leica DM5000B microscope (Germany) was used to examine the stained liver sections using a DCF 320 color camera and bright field settings. The fluorescent images were digitally captured using a DFC 350FX fluorescence camera.

## Fat red quantification

The fat red staining of the cryostat sections was assessed by bright-field microscopy. Digital images were captured from three contiguous microscopic fields per section, covering the entire parenchyma between large vessels. Using the UTHSCA Image Tool 3.0 software (University of Texas Health Science Centre, USA), the images were transformed to a binary format after appropriate thresholding. The same threshold was applied to all images from all sections. The fat red staining was quantified in the cultured hepatocytes as previously described (*Nunnari et al., 1989*). The values were normalized to amount of cellular protein quantified by Bradford assay (*Bradford, 1976*).

## Nile red staining

For Nile red staining, a 200 nM solution in PBS was prepared from a 1 mM stock solution in DMSO and added directly to the fixed cells. After 20 min incubation at room temperature, the cells were washed in PBS. The nuclei were counterstained with DAPI.

## Immunohistochemistry

Immunohistochemistry on paraffin sections (3 μm) was performed as previously described (*Zellmer et al., 2009*). The sections were boiled (3 × 5 min) in buffer (0.01 M sodium citrate, pH 6.0) and the slides were incubated for 30 min in 5% goat serum (Sigma-Aldrich, Germany) to block non-specific binding. The following antibodies were used: rabbit anti-FAS (1:400, Cell Signaling Technology, USA), rabbit anti-Cre (1:4000; Abcam, Cambridge), anti-GLI3 (1:1000; GeneTex, USA), anti-GLI1 (1:1000; GeneTex, USA), anti-SREBP1 (1:200; Abcam, Cambridge) anti-PPARG (1:250, Thermo-Fisher Scientific, Germany), biotinylated goat anti-rabbit IgG (Millipore, Germany) and Extravidine (Sigma-Aldrich, Germany). POD and counterstaining with hematoxylin were performed as previously described (*Zellmer et al., 2009*).

## Immunoblotting

In total the proteins from isolated hepatocytes from 9 SLC-WT and 9 SLC-KO mice were separated by 8% SDS-PAGE and transferred onto PVDF membranes (Roth, Germany), and incubated overnight at 4°C in blocking buffer (50 mM Tris HCl, pH 7.4, 150 mM NaCl, 0,1% Tween-20, 5% milkpowder). GLI3 antibody (GeneTex, USA) was diluted 1:20000 in Solution 1 from SignalBoost™ Immunoreaction Enhancer Kit (Calbiochem, Germany); ACTB antibody (Abcam, Cambridge) was diluted 1:5000 in 1% blocking buffer and incubated over night at 4°C. Blots were subsequently incubated with secondary anti rabbit POD antibody (Sigma, Germany). For the GLI3 Blots the secondary antibody was diluted 1:25000 in Solution 2 from the SignalBoost™ Immunoreaction Enhancer Kit. Chemiluminescent was used for detection. Densitometry quantification was performed with the Phoretix 1D Quantifier (Nonlinear dynamics, USA).

## Fatty acid and sterol biosynthesis

The biosynthesis of non-saponifiable lipids (sterols) and free fatty acids was performed as described (*Gebhardt et al., 2010*), with minor modifications. Briefly, the hepatocyte cultures were incubated in 1 ml of culture medium supplemented with 9 μM 1[14C]acetate (2.04 GBq/mmol; Amersham International) for 2 hr. Then, the cells were washed with 0.9% NaCl and lysed by incubation in 1 ml KOH (15% ) overnight at 37°C. The samples were saponified at 95°C for 90 min and the neutral lipids were extracted 3 times with a total volume of 8 ml of petroleum ether. The residual aqueous phase was acidified with 500 μl HCl (conc.), and the protonated fatty acids were extracted 3 times with a total volume of 8 ml of petroleum ether.

## Glycolysis

Glycolysis was determined as described by Probst et al. (*Probst et al., 1982*) with minor modifications. Briefly, the hepatocyte cultures were incubated in 1 ml of Hanks buffered solution supplemented with 20 mM NaHCO3, and 1.9 μM [14C(U)]glucose (9.7 GBq/mmol, Hartmann Analytic GmbH, Germany) for 2 hr. Then, 200 μl of the culture supernatant were applied to a Dowex 1X8 (formiate form) column and eluted with 7 ml sodium formiate (0.4 M). After adding Ultima-Gold™ AB solution (PerkinElmer GmbH, Germany), the radioactivity in each sample was counted in a liquid scintillation counter (Tri-carb 2500TR).

## Glycogen content

The glycogen determination was based on the microassay described by Gomez-Lechon et al. (*Gomez-Lechon et al., 1996*). Glycogen from bovine liver was used for calibration. The DNA of the cultured cells was measured by a fluorimetric method adapted to the 96-well format as previously described (*Gomez-Lechon et al., 1996*). The Hoechst 33,258 dye was replaced by DAPI (4',6-diamidino-2-phenylindole) at a concentration of 2.14 μg/ml in 40 mM Tris buffer (pH 7.0) containing 2 M NaCl and 1 mM EDTA. Fluorescence was measured using a microplate reader (Lumat LB 9501 Luminometer, Berthold, Germany). The protein concentrations were determined with the Bradford assay according to the standard microplate protocol (*Bradford, 1976*).

## ATP content

The hepatocytes were broken down using a TOS-UCD-200-EX Bioruptor to measure their ATP content. After centrifugation at 10,000 rpm at 4°C, the ATP content was determined using the CellTiter-Glo® Luminescent Cell Viability Assay according to the manufacturer's instructions. For normalization, the protein concentrations were determined using the Bradford assay according to the standard microplate protocol (*Bradford, 1976*).

## Activities of the mitochondrial respiratory complexes I-IV

The mitochondria were isolated from 10–25 mg of liver tissue to obtain a functional, purified, and intact mitochondrial fraction. The tissue was homogenized 3–4 times with a Teflon-on-glass Potter Elvehjem in 10 mM Tris-HCl (pH 7.8) buffer with 0.2 mM EDTA and 0.25 M sucrose. The activity of the respiratory chain complexes was assessed as described by Claus et al. (*Claus et al., 2013*).

## RNA preparation and quantitative real-time PCR (qRT-PCR)

After isolation of hepatocytes or tissue, the material was immediately frozen in liquid nitrogen and stored at -80°C up to 4–8 weeks. The total RNAs from the hepatocytes and tissues were extracted using TRIzol (VWR, Germany), and the RNeasy Lipid Tissue mini Kit (Qiagen, Germany) was used for the adipose tissue. The RNA was quantified by using a NanoTrop (VWR, Germany). 20 μl cDNA was prepared using 1 μg of RNA and oligo(dt) primers in combination with the IM Prom II reverse transcriptase (Promega, Germany) for each sample according to the manufacturer's instructions. For each gene, specific intron-spanning primers (*Supplementary file 1B*) were designed using the online tools Universal ProbeLibrary Probe-Finder software, Perl Primer and Primer 3. Therefore the RNA treatment with DNase was renounced. The levels of all mRNA transcripts were determined in duplicate by qRT-PCR using the LightCycler 2.0 Instrument and the LightCycler FastStart DNA Master PLUS SYBR Green I (Roche, Germany) according to the manufacturer's instructions. Using the standard curve method, the absolute amount of the specific PCR products for each primer set was quantified. *Actb* (beta Actin) was amplified from each sample for normalization as reference gene.

## RNA interference

The knockdown of *Gli1*, *Gli2* and *Gli3* in isolated hepatocytes from non-transgenic C57BL/6N mice, including the specific siRNA primers for the *Gli* factors, was performed as previously described (*Schmidt-Heck et al., 2015*). For *Sufu* knockdown, a specific siRNA and the respective nonsense oligo control siRNA were purchased from Invitrogen, Germany (*Supplementary file 1C*). The hepatocytes were seeded at a density of 100,000 cells per well on 12-well plates. After 4 h, the cells were transfected with the *Sufu* siRNA (10 nM) with INTERFERin (VWR, Germany) according to the manufacturer's instructions. 24 hr after transfection, the medium was changed, and fresh medium without

the siRNA was added. The changes in gene expression were analysed by qRT-PCR at different time points.

## GLI overexpression experiments

After isolation, hepatocytes from three male ob/ob mice at the age of 12 weeks were cultivated at 0.25 Mio. cells per well in 6-well plates in 1.5 ml medium (William's Medium E enriched with 10% fetal calf serum, 2 mM L-glutamine, 100 nM dexamethasone and Pen/Strep). After 2 h medium was changed Twenty four hours after seeding, transfection was performed using the jetPEI transfection reagent (VWR, Germany) according to the manufacturer's instruction with 1.0 µg DNA per well for each single plasmid. In the chase of the transfection of GLI1 and GLI3 together, the DNA amount of each plasmid was 0.5 µg DNA per well. For stable gene expression of *Gli1, Gli2* and *Gli3* the pFN1A HaloTag®CMV Flexi® Vector was used which was designed from Promega (Germany) in cooperation with the Kazusa DNA Research Institute (KDRI) (Japan) (*Nagase et al., 2008*). For control, only the DNA of the vector was transfected into the hepatocytes (MOCK transfection). RNA isolation and fat red quantification was performed 72 hr post transfection. Primers for human Gli expression are listed in *Supplementary file 1D*.

## Chromatin immunoprecipitation (ChIP)

To determine the binding of the transcription factor GLI3 to the promoter region of *Srebf1, Ppara* and *Fasn*, the Simple- ChIP Plus Enzymatic Chromatin IP Kit (Magnetic Beads) (Cell Signaling, Germany) was used according to the manufacturer' s instructions. Freshly isolated hepatocytes from three male C57BL/6N mice were pooled, washed and cross-linked with 37% formaldehyde. As positive experimental control the Histone H3 (D2B12) XP Rabbit mAb (#2729) was used. For negative control the normal Rabbit IgG was used. The cross-linked GLI3 DNA complex was precipitated with the goat anti mouse GLI3 antibody (20 µg) (R&D, Germany). The quantification analysis was performed using qRT-PCR with 2 µl of each DNA sample and specific primers listed in *Supplementary file 1E*. Primer pairs for the putative GLI3 binding site in the *Srebf1, Ppara* and *Fasn* promoter region were designed using Primer-BLAST software. The range of interest in the promoter regions (*Fasn*: ~ − 1767 bp; *Ppara*: -2674 bp; *Srebf1*: -1156 bp) was obtained by the MotifMap analysis (*Daily et al., 2011*). The primers for *Actb* were used to analyse the unspecific DNA-antibody binding. *Rpl30* primers (provided in the SimpleChIP Plus Enzymatic Chromatin IP Kit) were used as a positive control for the histone H3 antibody precipitation.

## Affymetrix chip

Microarray analysis was conducted using isolated hepatocytes from four independent SLC-WT and four SLC-KO mice, at the microarray core facility of the Interdisziplinäres Zentrum für klinische Forschung (IZKF) Leipzig (Faculty of Medicine, Leipzig University) described by Zellmer et al. (*Zellmer et al., 2009*). Briefly, GeneChip Mouse Genome 430 2.0 Arrays (Affymetrix) were used for hybridization. The gene expression data were pre-processed by Probe-level Linear Models using the 'affyPLM' packages of the Bioconductor Software.

To explore the biological function of differentially expressed genes (DEGs) the ClueGO (*Bindea et al., 2009*) Cytoscape plugin was applied to identify overrepresented GO categories and possible correlation between these categories. In addition, the ClueGO plugin was used to reduce redundancy by combining the significantly overrepresented categories to new groups.

The raw data for the micro array analysis is available at the temporary access link: http://seek.virtual-liver.de/data_files/3403?code=FC%2FJdE%2Ba94vTyN%2FcnSjG0uUL9elSQP4huKDrpFbV

## Protein–Protein Interaction (PPI) network

The STRING database version 10 (http://string-db.org) was used to explore the interactions between the studied genes/proteins related to steatosis and Hh signaling (*Szklarczyk et al., 2015*). A hypothetical network was constructed with four different data sources: known co-expression of the inputted proteins, experimental data, the results from various databases and a scan of published scientific abstracts. A high confidence score (0.7) was chosen.

## Statistical analysis

Values are expressed as means ± standard error of the mean (SEM). Statistical evaluations of data from different mice were made by unpaired t-test. For the RNAi experiments the paired student's t-test was used. The null hypothesis was rejected at the $p < 0.05$ (*); $p < 0.01$ (**) and $p < 0.001$ (***) levels.

# Acknowledgement

We thank Prof. Dr. Daniel Teupser and Franziska Jeromin for the serum measurements, Prof. Dr. Horst Robenek for the electron microscopy, and PD Dr. Knut Krohn for Affymetrix microarray abalysis. We cordially thank Kerstin Heise, Doris Mahn, Frank Struck, Fatina Siwczak and Vivien Karrasch for excellent technical assistance. Further, we would like to thank Petra Fink-Sterba, Sigrid Weisheit, Sandra Richter and Manuela Liebig from the MEZ (Faculty of Medicine) for taking excellent care of the mice. For help in questions around ChIP we would like to thank Stefanie Binder.

# Additional information

### Competing interests

MM-S: GeR is listed as inventor on a patent application filed by the University of Leipzig (WO 2013/110749A1, Therapeutic use of activators of zinc finger protein GLI3). The other authors declare that no competing interests exist.

### Funding

| Funder | Grant reference number | Author |
|---|---|---|
| Faculty of Medicine, Leipzig University | | Madlen Matz-Soja<br>Nora Klöting<br>Angela Schulz<br>Jürgen Kratzsch<br>Rolf Gebhardt |
| Bundesministerium für Bildung und Forschung | 0313081F | Jan Boettger<br>Wolfgang Schmidt-Heck<br>Sebastian Zellmer<br>Reinhardt Guthke<br>Rolf Gebhardt |
| Bundesministerium für Bildung und Forschung | 0315735 | Madlen Matz-Soja<br>Christiane Rennert<br>Kristin Schönefeld<br>Susanne Aleithe<br>Jan Boettger<br>Wolfgang Schmidt-Heck<br>Thomas S Weiss<br>Amalya Hovhannisyan<br>Sebastian Zellmer<br>Reinhardt Guthke<br>Rolf Gebhardt |
| Bundesministerium für Bildung und Forschung | 031L0053 | Madlen Matz-Soja<br>Christiane Rennert |
| Bundesministerium für Bildung und Forschung | FKZ 01E01501 | Nora Klöting |

The funders had no role in study design, data collection and interpretation, or the decision to submit the work for publication.

### Author contributions

MM-S, RGe, Conception and design, Acquisition of data, Analysis and interpretation of data, Drafting or revising the article, Contributed unpublished essential data or reagents; CR, Acquisition of data, Analysis and interpretation of data, Drafting or revising the article, Contributed unpublished essential data or reagents; KS, SA, AH, JK, Acquisition of data, Analysis and interpretation of data; JB, Acquisition of data, Analysis and interpretation of data, Contributed unpublished essential data

or reagents; WS-H, RGu, Analysis and interpretation of data, Drafting or revising the article; TSW, Provided the human material, Drafting or revising the article; SZ, Conception and design, Acquisition of data, Analysis and interpretation of data, Drafting or revising the article; NK, Provided the ob/ob mice, Drafting or revising the article; AS, Provided the MC4R mice., Analysis and interpretation of data, Drafting or revising the article

## Author ORCIDs
Madlen Matz-Soja, ID http://orcid.org/0000-0002-5929-3704

## Ethics
Human subjects: Experimental procedures were performed according to the guidelines of the charitable state controlled foundation Human Tissue and Cell Research (HTCR), with the written informed patient's consent approved by the local ethical committee of the University of Regensburg (12-101-0048).

Animal experimentation: The mice were maintained in a pathogen-free facility on a 12:12 h LD cycle, according to the German guidelines and the world medical association declaration of Helsinki for the care and safe use of experimental animals.

## Additional files

### Supplementary files
• Supplementary file 1. (A) Primers for genotyping the SLC mice. (B) Primers for qRT-PCR on murine mRNA. (C) Primers for siRNA mediated knockdown of *Sufu*. (D) Primers for ChIP (E) Primers for human Gli factors

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
