## [Decision Letter]

Thank you for submitting your work entitled "Liver steatosis – a new playground for Hedgehog signaling in mammals" for consideration by *eLife*. Your article has been favorably evaluated by Naama Barkai (Senior editor) and two reviewers, one of whom is a member of our Board of Reviewing Editors.

The reviewers have discussed the reviews with one another and the Reviewing Editor has drafted this decision to help you prepare a revised submission.

Your manuscript reports unexpected effects of the Hedgehog pathway on liver steatosis in mouse and human tissues and appears to represent an interesting and important advance in the field. You show attenuation of the Hedgehog pathway in mouse liver, causing enhanced lipid accumulation, appears to be associated with altered abundance of the transcription factors GLI1 and GLI3. In turn, the "steatotic Gli-code" operates by modulating sets of other transcription factors and target enzymes to cause lipid synthesis.

The major problem raised by the reviewers is the lack of mechanistic data addressing the specific pathways by which the Gli factors actually modulate the downstream transcription factors purported to mediate the effects observed. Have ChIP experiments been performed to assess this issue? Related to this, and equally important, what are the mechanisms by which the specific lipogenic enzymes are regulated? For example, are SREBP or ChREBP or both (or others?) mediating the regulation of FASN? These mechanistic gaps in the study are of major importance in assessing impact of the study, which at this point falls short of publication. Such mechanistic data is considered necessary for publication, and we are hopeful you will be able to provide a revised version of the paper that includes such data.

Other issues that should be directly addressed in a revised manuscript include: is insulin resistance associated with the steatosis in SLC-KO mice and what are the parameters of insulin signaling in this model? Also, protein contents of *Gli1* and *Gli3* as well as ChREBP and SREBP should be provided since the mRNA values may not reflect the actual machinery that modulates the observed effects. Is *Gli3* overexpression in ob/ob liver sufficient to overcome the steatosis?

---

## [Author Response]

The major problem raised by the reviewers is the lack of mechanistic data addressing the specific pathways by which the Gli factors actually modulate the downstream transcription factors purported to mediate the effects observed. Have ChIP experiments been performed to assess this issue?

We fully agree that mechanistic data addressing how the Gli factors modulate the downstream transcription factors to mediate the effects of Hh signalling observed in our study is scarce. So far, however, this is an unsatisfying situation for GLI-dependent transcriptional regulation in general. In part, this is due to the enormous complexity of this kind of regulation, involving not only two different forms – a full-length activator and a truncated repressor form in the case of GLI2 and GLI3, and an activator form only in case of GLI1 –, but also different phosphorylated states of all these forms. An additional drawback is the lack of suitable tools, e. g. the lack of reliable antibodies for distinguishing these different molecular entities.

Despite these unfavourable conditions, we have taken the request of the reviewers as a motivation for performing a ChIP experiment with GLI3 according to what we have published for the regulation of *Igf1* expression by GLI3 (Matz-Soja et al. 2014). The results of these experiments are now provided in Figure 7—figure supplement 2 and give a first hint that direct binding of GLI3 to a binding site predicted by MotifMap in the promoter region of Srebf1 (at -1156 bp) might be involved in transmitting GLI factor activity to the expression of SREBP1. No such hint was found, however, in the case of *Fasn* (at the predicted binding site at -1767 bp) and of *Ppara* (at the predicted binding site at -2674 bp)suggesting that these genes are not primary, but secondary targets of GLI3 (mediated among else by SREBP1). Fortunately, our results are confirmed by recent work of Gurdziel and coworkers (Gurdziel et al. 2016) published during revision of our manuscript, who used RNA sequencing and other techniques to show enriched GLI binding to binding motifs in the promotor regions of (among else) *Srebf1* and *Pnpla3.*

Although all this data is valuable information favoring a direct regulation of *Srebf1* (and possibly of other lipogenic TFs) by GLI3, we are aware that additional work is needed to corroborate these findings and to clearly differentiate them from an indirect one via, for instance, GATA4 as suggested by Suh et al. (Suh et al. 2006) for adipocytes. However, we hope the reviewers will understand that such work cannot be done as a simple addendum to the present study.

The results of the chromatin immunoprecipitation (ChIP) experiments are now provided in Figure 7—figure supplement 2. Accordingly, technical details were given in the Methods section and the results were presented in the text and discussed. Furthermore, we cited the confirming work of Gurdziel et al. and added this paper to the reference list.

*Related to this, and equally important, what are the mechanisms by which the specific lipogenic enzymes are regulated? For example, are SREBP or ChREBP or both (or others?) mediating the regulation of FASN? These mechanistic gaps in the study are of major importance in assessing impact of the study, which at this point falls short of publication. Such mechanistic data is considered necessary for publication, and we are hopeful you will be able to provide a revised version of the paper that includes such data.* We apologize for not having addressed this point in the original manuscript with the necessary care. Unfortunately, we cited only one (Reed et al. 2008) of the many studies that have demonstrated in detail that important enzymatic genes of liver lipid metabolism (e.g. *Fasn*) are regulated via transcription factors like SREBP1, ChREBP and others. For instance, it was already shown that SREBP1 has binding sites at -7000 bp and -500 bp of the *Fasn* gene (Amemiya-Kudo et al.; Morishita et al. 2014). Furthermore it was also shown that overexpression of SREBP 1c, an isoform of SREBP1, lead to the significant upregulation of *Fasn* gene expression in hepatocytes (Dentin et al. 2004). Obviously, SREBP1 does not act alone but in combination with several other TFs such as the NFYs and SP1 (Reed et al. 2008). For ChREBP the situation is similar (Morishita et al. 2014). Apart from these major publications a lot of additional evidence in favor of such regulation was published by many other groups (reviewed in (Postic und Girard 2008; Morishita et al. 2014), although mainly in adipose tissue. Thus, we think that giving enough credit to this work in our manuscript should sufficiently answer the point raised by the reviewers even for the specific case of the liver.

Furthermore, we have discussed in more detail that we do not consider ChREBP as a direct target of the GLI factors, because it is changed in the SLC-KO mice, but not in response to the siRNA interference of each single Gli transcription factor. In contrast, we assume an indirect regulation that deserves thorough investigation in the future. An attractive candidate for a mediating signal is the IGF axis which we found to be directly influenced by the *Smo* knockout in SLC-KO mice (Matz-Soja et al. 2014).

Concerning the regulation of lipogenic enzymes, references to Amemiya-Kudo et al.; Morishita et al. 2014; Dentin et al. 2004 were added to the Discussion and the reference list. The specific situation of ChREBP was discussed in more detail.

Other issues that should be directly addressed in a revised manuscript include: is insulin resistance associated with the steatosis in SLC-KO mice and what are the parameters of insulin signaling in this model?

The question of whether insulin resistance is associated with steatosis in SLC-KO mice is highly relevant. However, from the fact that the serum concentrations of insulin in SLC mice obviously were not changed due to the Smo-KO (see Table 1), we concluded that insulin resistance is not associated with the steatosis in SLC-KO mice. This view is further supported by additional parameters of insulin signalling in the SLC model. For instance, we have measured the expression of insulin receptor (*Inrs*), insulin receptor substrate 1 (*Irs1*) and insulin receptor substrate 2 (*Irs2*) via qRT-PCR and found no significant differences. Furthermore, we have performed GTT (glucose tolerance tests) and an ITT (insulin tolerance test) in male SAC mice (a slightly different model to SLC-KO mice) and found no differences between KO and WT animals. Collectively, these results make it rather unlikely that the SLC-KO mice suffer from insulin resistance, at least during the first 5 weeks after *Smo* knockout.

The results of the qRT-PCR were added to Figure 2—figure supplement 1. Accordingly, they were shortly mentioned in the text.

Also, protein contents of Gli1 and Gli3 as well as ChREBP and SREBP should be provided since the mRNA values may not reflect the actual machinery that modulates the observed effects.

We agree that mRNA levels alone cannot reflect the actual machinery that modulates the observed effects in the transgenic mice. Therefore, we have determined the protein content of GLI3 by western blotting and show that it is significantly reduced in response to Smo KO. To visualize the protein amount and particularly the distribution of the GLI1 and GLI3 protein in the liver parenchyma of the SLC mice, we performed immunohistological stainings of GLI1 and GLI3 as well. This is important, since Hh signalling is heterogeneously distributed in liver parenchyma (Gebhardt und Matz-Soja 2014)(Gebhardt und Matz-Soja 2014). The results clearly demonstrate that both transcription factors, GLI1 and GLI3, are well detectable in hepatocyte nuclei in SLC-WT animals, but are absent in nuclei of SLC-KO hepatocytes. In SLC-KO livers, these TFs are only present in non-parenchymal cells, e.g. bile duct epithelial cells.

Similar stainings were performed for the transcription-factors SREBP1 and PPARG. SREBP1 protein showed a slightly pericentral and rare nuclear preference in liver parenchyma from SLC-WT mice and was very strongly enhanced in the pericentral zone of livers from SLC-KO mice. In particular, the frequency of nuclear staining in pericentral hepatocytes was much higher in these mice compared to control mice.

Likewise, PPARG protein was much more frequent in hepatocyte nuclei of SLC-KO mice than in SLC-WT mice. Also with PPARG a heterogeneous distribution of the protein was obvious.

In essence, these results demonstrate that the changes seen in the amount of protein for all these TFs reflect those of the mRNA levels. Moreover, they provide valuable additional information about the zonal expression and the functional characteristic (e. g. nuclear localization) of the GLI factors and the lipogenic TFs.

We have now added the protein content of GLI3 determined by western blotting to the manuscript in Figure 3—figure supplement 1. The results of the immunohistological stainings of GLI1 and GLI3 were added to Figure 3. The results for protein expression of the transcription-factors SREBP1 and PPARG were combined in a new Figure 4. Accordingly, all the additional results were described and discussed in the text.

Is Gli3 overexpression in ob/ob liver sufficient to overcome the steatosis?

We agree that the overcoming of the steatosis in ob/ob mice via GLI overexpression is very interesting and challenging because of the completely different mechanism by which the steatosis is induced in ob/ob mice.

We have performed GLI overexpression experiments in isolated hepatocytes from three male ob/ob mice. The cells were transfected either separately with each of the appropriate plasmids alone or with a combination of the plasmids for GLI1 and GLI3. 72 h post transfection we quantified the mRNA expression of human *Gli1, Gli2* and *Gli3* and performed fat red staining to depict and measure the changes in lipid content. Interestingly, our results demonstrate that *only* the combined overexpression of GLI1 and GLI3 (but not of each Gli factor alone) is able to significantly reduce the lipid content in the hepatocytes from ob/ob mice (Figure 9). These findings emphasize the strong influence of the GLI-code and, thus, the mediating function of Hh signaling in an animal model of steatosis of completely different origin (i.e. leptin deficiency).

The results are shown in the new Figure 9. Accordingly, all the additional results were described and discussed in the text.